# Confidence intervals for rainfall dispersions using the ratio of two coefficients of variation of lognormal distributions with excess zeros

**Noppadon Yosboonruang**, **Sa-Aat Niwitpong** \*, **Suparat Niwitpong**

Department of Applied Statistics, Faculty of Applied Science, King Mongkut's University of Technology North Bangkok, Bangkok, Thailand

These authors contributed equally to this work.

\* sa-aat.n@sci.kmutnb.ac.th

**Data Availability Statement:** All relevant data are within the paper.

**Funding:** The authors gratefully acknowledge King Mongkut's University of Technology North

## Abstract

Rainfall fluctuation is directly affected by the Earth's climate change. It can be described using the coefficient of variation (CV). Similarly, the ratio of CVs can be used to compare the rainfall variation between two regions. The ratio of CVs has been widely used in statistical inference in a number of applications. Meanwhile, the confidence interval constructed with this statistic is also of interest. In this paper, confidence intervals for the ratio of two independent CVs of lognormal distributions with excess zeros using the fiducial generalized confidence interval (FGCI), Bayesian methods based on the left-invariant Jeffreys, Jeffreys rule, and uniform priors, and the Wald and Fieller log-likelihood methods are proposed. The results of a simulation study reveal that the highest posterior density (HPD) Bayesian using the Jeffreys rule prior method performed the best in terms of the coverage probability and the average length for almost all cases of small sample size and a large sample size together with a large variance and a small proportion of non-zero values. The performance of the statistic is demonstrated on two rainfall datasets from the central and southern regions in Thailand.

## Introduction

The Earth's climate is changing due to increased greenhouse gas emission from human activities, and climate change has resulted in dramatic weather events such as heatwaves, heavy rainfall, droughts, etc. Thailand is a country situated in the southeastern region of Asia that is affected by the southwest and northeast monsoons at different times of the year [1]. The country is divided into five regions: North, Northeast, Central, East, and South. Over the past three decades, Thailand has suffered from increased temperatures and fluctuating rainfall. In this study, rainfall is of interest because too much rain causes flooding and too little causes droughts at different times of the year in each area of the country. Rainfall fluctuation can be described using the CV, meaning that the ratio of CVs can be used to compare the rainfall variation between two regions. The lognormal distribution with excess zeros, a mixed distribution of discrete and continuous random variables, has been applied in many studies involving

Bangkok. Contract no. KMUTNB-65-KNOW-09. The funders had no role in study design, data collection and analysis, decision to publish, or preparation of the manuscript.

**Competing interests:** The authors have declared that no competing interests exist.

observations with zero values (such as rainfall data) [2–6]. Observations with zero values conform to a binomial distribution whereas the positive values with skewness follow a lognormal distribution. Moreover, the lognormal distribution with excess zeros has been widely used in fields such as fishery surveys [7–9], climatology [10, 11], and medicine [12, 13].

The CV is the ratio of the standard deviation to the mean that is more widely used than the standard deviation when comparing the dispersion in two or more datasets. There have been many applications of the CV to compare the dispersion in datasets. For example, Verrill and Johnson [14] compared the CVs of the xylan percentage for laboratory quality control data in the United States Department of Agriculture forest products laboratory, while Gulhar et al. [15] established confidence intervals for health-related datasets, including birth weight and cigarette smoking. Nam and Kwon [16] applied CVs to estrogen metabolites in blood and urine measurement data. Marek [17] used CVs for mining and ore geology to classify bituminous coal deposits in the Upper Silesian Coal Basin in Poland.

In statistical inference, the confidence intervals for the CV and the function of CVs have attracted the interest of several researchers. Wong and Wu [18] recommended a small sample asymptotic method to construct the confidence intervals for the CV of small sample sizes following normal, gamma, or Weibull distributions. Mahmoudvand and Hassani [19] proposed two confidence intervals for the CV for a normal distribution based on an asymptotically unbiased estimator for the CV that worked well, especially for small sample sizes. Hayter [20] applied confidence intervals (a directional bound and a non-directional upper bound) for the CV of a normal distribution applied to win probabilities. For the ratio of CVs, Nam and Kwon [16] introduced confidence intervals based on the Wald-type, Fieller-type, and log methods, and the method of variance estimate recovery (MOVER) of lognormal distributions. Hasan and Krishnamoorthy [21] suggested MOVER and the fiducial method to establish confidence intervals for small sample sizes from lognormal distributions. Wong and Jiang [22] proposed the Bartlett-corrected likelihood ratio method to construct confidence intervals for lognormal distributions. In addition, Sangnawakij and Niwitpong [23] used the score and Wald methods to construct confidence intervals for the CV and function of CVs with bounded parameter spaces in two gamma distributions for which both proposed methods were evidently suitable. They also constructed confidence intervals using MOVER, the generalized confidence interval (GCI), and the asymptotic confidence interval for CV and the difference between CVs with two parameters exponential distributions of which the GCI was satisfied [24]. Moreover, for a lognormal distribution with excess zeros, Buntao and Niwitpong [25] presented the generalized pivotal approach (GPA) and the closed-form method for obtaining the confidence intervals for the difference between CVs and reported that GPA performed well. Later, the same authors established the confidence intervals for the ratio of CVs under the GPA and MOVER methods, the results revealing that GPA was the most accurate [26]. Recently, Yosboonruang et al. [27] compared GCI and a modified Fletcher method in the confidence interval construction for CV (GCI was the most suitable). After that, they suggested FGCI, comparable with MOVER, to establish the confidence intervals for the CV of a lognormal distribution with excess zeros with three parameters [28]. Moreover, Yosboonruang et al. [29] reported a Bayesian confidence interval compared with FGCI, the results indicating that the Bayesian method performed the best.

In this paper, the ratio of CVs from two lognormal distributions with excess zeros is proposed. Therefore, the concept of FGCI [28], the Bayesian method [29], Wald- and Fieller-type methods [16] were extended to establish confidence intervals for the ratio of two independent CVs from lognormal distributions with excess zeros. The proposed methods for constructing the confidence intervals are presented in the next section. Subsequently, the results of a simulation study are reported to assess the coverage probabilities and average lengths for comparing

the proposed methods. Next, application of the proposed methods to rainfall datasets from two regions in Thailand is demonstrated. Last, the paper is brought to a close with a discussion and conclusion.

## Materials and methods

Let $\mathbf{X}_{ij} = (X_{i1}, X_{i2}, \ldots, X_{in_i})$, for $i = 1, 2, j = 1, 2, \ldots, n_i$, be a semi-continuous random sample that conforms a lognormal distribution with excess zeros with the probability of zero values $\delta_{i,0}$, mean $\mu_i$, and variance $\sigma_i^2$, denoted by $X_{ij} \sim \Delta(\delta_{i,0}, \mu_i, \sigma_i^2)$. The zero observations have a binomial distribution, while the non-zero observations follow a lognormal distribution. The numbers of zero and non-zero observations are defined as $n_{i,0}$ and $n_{i,1}$, respectively, where $n_i = n_{i,0} + n_{i,1}$. This leads to the distribution function of $X_{ij}$:

$$F\left(x_{ij;\delta_{i,0},\mu_i,\sigma_i^2}\right) = \begin{cases} \delta_{i,0} & ; x_{ij} = 0 \\ \delta_{i,0} + (1 - \delta_{i,0}) H(x_{ij}; \mu_i, \sigma_i^2) & ; x_{ij} > 0, \end{cases} \tag{1}$$

where $\delta_{i,0} = P(x_{ij} = 0)$, $n_{i,0} \sim B(n_i, \delta_{i,0})$ [6], and $H(x_{ij}; \mu_i, \sigma_i^2)$ is a lognormal cumulative distribution function [11], so $\ln X_{ij}$ follows a normal distribution with mean $\mu_i$ and variance $\sigma_i^2$ for $X_{ij} > 0$. Thus, the probability density function of $X_{ij}$ can be expressed as

$$f(x_{ij}; \delta_{i,0}, \mu_i, \sigma_i^2)$$

$$= \delta_{i,0} I_0\left[x_{ij}\right] + \left(1 - \delta_{i,0}\right) \frac{1}{x_{ij}\sqrt{2\pi}\sigma_i} \exp\left[-\frac{1}{2}\left(\frac{\ln x_{ij} - \mu_i}{\sigma_i}\right)^2\right] I_{(0,\infty)}\left[x_{ij}\right] \tag{2}$$

such that if $x_{ij} = 0$, then $I_0[x_{ij}] = 1$ and $I_{(0,\infty)}[x_{ij}] = 0$, and if $x_{ij} > 0$, then $I_{(0,\infty)}[x_{ij}] = 1$. According to Aitchison [30], the population mean and variance of $X_{ij}$ are $\mu_{X_{ij}} = \delta_{i,1} \exp\left(\mu_i + \sigma_i^2/2\right)$ and $\sigma_{X_{ij}}^2 = \delta_{i,1} \exp\left(2\mu_i + \sigma_i^2\right)[\exp\left(\sigma_i^2\right) - \delta_{i,1}]$, respectively, where $\delta_{i,1} = 1 - \delta_{i,0}$. Thus, the CV of $X_{ij}$ can be defined as

$$CV\left(X_{ij}\right) = \eta_i = \sqrt{\frac{\exp\left(\sigma_i^2\right) - \delta_{i,1}}{\delta_{i,1}}}. \tag{3}$$

The aim here is to construct the confidence interval for the ratio of the CVs:

$$\phi = \frac{\sqrt{\dfrac{\exp(\sigma_1^2) - \delta_{1,1}}{\delta_{1,1}}}}{\sqrt{\dfrac{\exp(\sigma_2^2) - \delta_{2,1}}{\delta_{2,1}}}}. \tag{4}$$

In accordance with a lognormal distribution with excess zeros, the maximum likelihood estimators (MLEs) of parameters $\delta_{i,1}$, $\delta_{i,0}$, and $\mu_i$ are $\widehat{\delta}_{i,1} = n_{i,1}/n_i$, $\widehat{\delta}_{i,0} = n_{i,0}/n_i$, and $\widehat{\mu}_i = \sum_{i=1}^{n_{i,1}} \ln x_{ij}/n_{i,1}$, respectively. For $\sigma_i^2$, the unbiased estimator is $\widehat{\sigma}_i^2 = \sum_{i=1}^{n_{i,1}} (\ln x_{ij} - \widehat{\mu}_i)^2/(n_{i,1} - 1)$. The approaches used to construct the confidence intervals are in the following subsections.

## The fiducial generalized confidence interval

Fiducial inference was initially suggested by Fisher [31], whereupon Hannig [32] and Li, Zhou, and Tian [33] introduced the generalized fiducial quantities (GFQs) for $\delta_{i,1}$ and $\sigma_i^2$, which are respectively

$$R_{\delta_{i,1}} \sim \frac{1}{2} Beta\left(n_{i,1}, n_{i,0} + 1\right) + \frac{1}{2} Beta\left(n_{i,1} + 1, n_{i,0}\right) \tag{5}$$

and

$$R_{\sigma_i^2} = \frac{(n_{i,1} - 1)\widehat{\sigma}_i^2}{U_i}, \tag{6}$$

where $U_i \sim \chi_{n_{i,1}-1}^2$.

Moreover, there have been several studies using FGCI to construct confidence intervals [28, 29, 32, 34–37]. From Eqs (5) and (6), the GFQ for $\eta_i$ is defined as

$$R_{\eta_i} = \sqrt{\frac{\exp\left(R_{\sigma_i^2}\right) - R_{\delta_{i,1}}}{R_{\delta_{i,1}}}}. \tag{7}$$

The approach in this study points toward constructing the confidence interval for the ratio of CVs. Thus, the GFQ for $\phi$ is in the form

$$R_\phi = \frac{\sqrt{\dfrac{\exp(R_{\sigma_1^2}) - R_{\delta_{1,1}}}{R_{\delta_{1,1}}}}}{\sqrt{\dfrac{\exp(R_{\sigma_2^2}) - R_{\delta_{2,1}}}{R_{\delta_{2,1}}}}}. \tag{8}$$

Therefore, the $100(1 - \alpha)\%$ confidence interval for $\phi$ is

$$\text{CI}_\phi^{\text{FGCI}} = [R_{\phi,l}, R_{\phi,u}] = [R_\phi(\alpha/2), R_\phi(1 - \alpha/2)], \tag{9}$$

where $R_\phi(\alpha/2)$ and $R_\phi(1 - \alpha/2)$ are the $(\alpha/2)$-th and $(1 - \alpha/2)$-th percentiles of $R_\phi$, respectively.

**Algorithm 1**

```
for k = 1 : M do
  Generate datasets xᵢⱼ, for i = 1, 2, j = 1, 2, ..., nᵢ, from the log-
  normal distributions with excess zeros;
  Calculate δ̂ᵢ,₁ and σ̂ᵢ²;
  for l = 1 : m do
    Generate Beta (nᵢ,₁, nᵢ,₀ + 1) and Beta (nᵢ,₁ + 1, nᵢ,₀);
    Calculate R_δᵢ,₁, R_σᵢ², and R_φ;
  end for
  Calculate the (α/2)-th and (1 − α/2)-th percentiles of R_φ
end for
```

## Bayesian methods

The probability density function of a lognormal distribution with excess zeros (Eq (2)) has unknown parameters $\delta_{i,0}$, $\mu_i$ and $\sigma_i^2$. The joint likelihood function is defined as

$$L\left(\delta_{i,0}, \mu_i, \sigma_i^2 \mid \mathbf{x}_{ij}\right) \propto \prod_{i=1}^2 \left\{ \delta_{i,0}^{n_{i,0}} (1 - \delta_{i,0})^{n_{i,1}} \prod_{j=1}^{n_{i,1}} \frac{1}{\sigma_i} \exp\left[ -\frac{1}{2\sigma_i^2} (\ln x_{ij} - \mu_i)^2 \right] \right\}. \tag{10}$$

Since we are interested in the ratio of the CVs, then the Fisher information matrix of the unknown parameters $(\delta_{1,0}, \mu_1, \sigma_1^2, \delta_{2,0}, \mu_2, \sigma_2^2)$ computed by the second-order derivative of the log-likelihood function can be expressed as

$$
I(\delta_{1,0}, \mu_1, \sigma_1^2, \delta_{2,0}, \mu_2, \sigma_2^2)
$$

$$
= \text{diag} \left[ \frac{n_1}{\delta_{1,0}(1 - \delta_{1,0})} \quad \frac{n_{1,1}}{\sigma_1^2} \quad \frac{n_1(1 - \delta_{1,0})}{2(\sigma_1^2)^2} \quad \frac{n_2}{\delta_{2,0}(1 - \delta_{2,0})} \quad \frac{n_{2,1}}{\sigma_2^2} \quad \frac{n_2(1 - \delta_{2,0})}{2(\sigma_2^2)^2} \right]. \tag{11}
$$

Subsequently, the equitailed confidence intervals and the HPD intervals are constructed for the left-invariant Jeffreys, the Jeffreys rule, and the uniform priors.

**The left-invariant Jeffreys prior.** Because the lognormal distribution with excess zeros is a combination of binomial and lognormal distributions, the Jeffreys priors for $\delta_{i,0}$ and $\sigma_i^2$ are computed under these distributions. According to Jeffreys [38] and Ghosh et al. [39], the invariant Jeffreys prior is obtained using a Fisher information matrix ($I(\theta)$), which is given as $p(\theta) = \sqrt{|I(\theta)|}$.

Because the left-invariant Jeffreys prior is non-informative, the Jeffreys invariant prior for a binomial proportion ($\delta_{i,0}$) is given by

$$
p(\delta_{i,0}) = \sqrt{|I(\delta)|} \propto \delta_{i,0}^{-\frac{1}{2}}(1 - \delta_{i,0})^{-\frac{1}{2}}. \tag{12}
$$

This leads to the posterior distribution of $\delta_{i,0}$ as

$$
p(\delta_{i,0} \mid n_{i,0}) \propto \delta_{i,0}^{n_{i,0}-\frac{1}{2}}(1 - \delta_{i,0})^{n_{i,1}-\frac{1}{2}}, \tag{13}
$$

which is a beta distribution with parameters $n_{i,0} + 1/2$ and $n_{i,1} + 1/2$, denoted by $\delta_{i,0} \mid n_{i,0} \sim$ *Beta* $(n_{i,0} + 1/2, n_{i,1} + 1/2)$. Similarly for the lognormal distribution, the left-invariant Jeffreys prior for $\sigma_i^2$, which is $p(\sigma_i^2) = 1/\sigma_i^2$ [39]. Subsequently, by combining the prior distributions of $\delta_{i,0}$ and $\sigma_i^2$, we obtain $p(\delta_{i,0}, \sigma_i^2) \propto \sigma_i^{-2}\delta_{i,0}^{-\frac{1}{2}}(1 - \delta_{i,0})^{-\frac{1}{2}}$ for a lognormal distribution with excess zeros. Accordingly, the joint posterior density function for a lognormal distribution with excess zeros is written as

$$
p(\delta_{i,0}, \sigma_i^2 \mid x_{ij}) = \prod_{i=1}^{2} \left\{ \frac{1}{Beta\left(n_{i,0} + \frac{1}{2}, n_{i,1} + \frac{1}{2}\right)} \delta_{i,0}^{n_{i,0}-\frac{1}{2}}(1 - \delta_{i,0})^{n_{i,1}-\frac{1}{2}} \frac{1}{\sqrt{2\pi} \frac{\sigma_i}{\sqrt{n_{i,1}}}} \right.
$$

$$
\times \exp\left[ -\frac{1}{2\frac{\sigma_i^2}{n_{i,1}}}(\mu_i - \hat{\mu}_i)^2 \right] \frac{\left[\frac{(n_{i,1}-1)\hat{\sigma}_i^2}{2}\right]^{\frac{n_{i,1}-1}{2}}}{\Gamma\left(\frac{n_{i,1}-1}{2}\right)}(\sigma_i^2)^{-1-\frac{n_{i,1}-1}{2}} \tag{14}
$$

$$
\left. \times \exp\left[ -\frac{(n_{i,1}-1)\hat{\sigma}_i^2}{2\sigma_i^2} \right] \right\},
$$

where $\hat{\mu}_i = \sum_{j=1}^{n_{i,1}} \ln x_{ij}/n_{i,1}$ and $\hat{\sigma}_i^2 = \sum_{j=1}^{n_{i,1}} (\ln x_{ij} - \hat{\mu}_i)^2/(n_{i,1} - 1)$. Therefore, the posterior distribution of $\delta_{i,0}$ is a beta distribution with parameters $n_{i,0} + 1/2$ and $n_{i,1} + 1/2$, denoted by $\delta_{i,0}|x_{ij} \sim Beta(n_{i,0} + 1/2, n_{i,1} + 1/2)$. Similarly, the posterior distribution of $\sigma_i^2$ is an inverse gamma distribution with parameters $(n_{i,1} - 1)/2$ and $(n_{i,1} - 1)\hat{\sigma}_i^2/2$, denoted by $\sigma_i^2 \mid x_{ij} \sim IG[(n_{i,1} - 1)/2, (n_{i,1} - 1)\hat{\sigma}_i^2/2]$.

**The Jeffreys rule prior.** The Jeffreys rule prior, which was previously referred to as the square root of the determinant of the Fisher information for a binomial proportion, is $p(\delta_{i,0}) \propto \delta_{i,0}^{-\frac{1}{2}}(1 - \delta_{i,0})^{\frac{1}{2}}$ and for $\sigma_i^2$ from a lognormal distribution is $p(\sigma_i^2) \propto \sigma_i^{-3}$ [40]. According to the CV of the lognormal distribution with excess zeros as in Eq (3), the parameters are independent, then the Jeffreys rule prior for $(\delta_{i,0}, \sigma_i^2)$ is $p(\delta_{i,0}, \sigma_i^2) \propto \sigma_i^{-3} \delta_{i,0}^{-\frac{1}{2}}(1 - \delta_{i,0})^{\frac{1}{2}}$. Therefore, the joint posterior density function is defined as

$$p(\delta_{i,0}, \sigma_i^2 \mid x_{ij}) =$$

$$\prod_{i=1}^{2}\left\{ \frac{1}{Beta\left(n_{i,0} + \frac{1}{2}, n_{i,1} + \frac{3}{2}\right)} \delta_{i,0}^{n_{i,0}-\frac{1}{2}}(1 - \delta_{i,0})^{n_{i,1}+\frac{1}{2}} \frac{1}{\sqrt{2\pi}\frac{\sigma_i}{\sqrt{n_{i,1}}}} \right.$$

$$\left. \times \exp\left[ -\frac{1}{2\frac{\sigma_i^2}{n_{i,1}}}(\mu_i - \widehat{\mu}_i)^2 \right] \frac{\left[\frac{n_{i,1}\widehat{\sigma}_i^2}{2}\right]^{\frac{n_{i,1}}{2}}}{\Gamma\left(\frac{n_{i,1}}{2}\right)}(\sigma_i^2)^{-1-\frac{n_{i,1}}{2}} \exp\left[ -\frac{n_{i,1}\widehat{\sigma}_i^2}{2\sigma_i^2} \right] \right\},$$

(15)

where $\widehat{\mu}_i = \sum_{j=1}^{n_{i,1}} \ln x_{ij}/n_{i,1}$ and $\widehat{\sigma}_i^2 = \sum_{j=1}^{n_{i,1}}(\ln x_{ij} - \widehat{\mu}_i)^2/(n_{i,1} - 1)$. Subsequently, the posterior densities of $\delta_{i,0}$ and $\sigma_i^2$ follow a beta distribution, $Beta(n_{i,0} + 1/2, n_{i,1} + 3/2)$, and an inverse gamma distribution, $IG(n_{i,1}/2, n_{i,1}\widehat{\sigma}_i^2/2)$, respectively.

**The uniform prior.** For the uniform prior, the prior probability is a constant function whereby all possible values are equally likely to be a *priori* [41, 42]. Accordingly, for binomial and lognormal distributions, the uniform priors of $\delta_{i,0}$ and $\sigma_i^2$ are proportional to 1 [43, 44], which implies that the uniform prior for a lognormal distribution with excess zeros is $p(\delta_{i,0}, \sigma_i^2) \propto 1$. The joint posterior distribution for a lognormal distribution with excess zeros is given by

$$p(\delta_{i,0}, \sigma_i^2 \mid x_{ij}) = \prod_{i=1}^{2}\left\{ \frac{1}{Beta(n_{i,0} + 1, n_{i,1} + 1)} \delta_{i,0}^{n_{i,0}}(1 - \delta_{i,0})^{n_{i,1}} \frac{1}{\sqrt{2\pi}\frac{\sigma_i}{\sqrt{n_{i,1}}}} \right.$$

$$\times \exp\left[ -\frac{1}{2\frac{\sigma_i^2}{n_{i,1}}}(\mu_i - \widehat{\mu}_i)^2 \right] \frac{\left[\frac{(n_{i,1} - 2)\widehat{\sigma}_i^2}{2}\right]^{\frac{n_{i,1}-2}{2}}}{\Gamma\left(\frac{n_{i,1} - 2}{2}\right)}(\sigma_i^2)^{-1-\frac{n_{i,1}-2}{2}}$$

(16)

$$\left. \times \exp\left[ -\frac{(n_{i,1} - 2)\widehat{\sigma}_i^2}{2\sigma_i^2} \right] \right\},$$

where $\widehat{\mu}_i = \sum_{j=1}^{n_{i,1}} \ln x_{ij}/n_{i,1}$ and $\widehat{\sigma}_i^2 = \sum_{j=1}^{n_{i,1}}(\ln x_{ij} - \widehat{\mu}_i)^2/(n_{i,1} - 1)$. Thus, the posterior distribution for $\delta_{i,0}$ follows a beta distribution, $Beta(n_{i,0} + 1, n_{i,1} + 1)$, and that of $\sigma_i^2$ is an inverse gamma distribution, $\sigma_i^2 \mid x_{ij} \sim IG[(n_{i,1} - 2)/2, (n_{i,1} - 2)\widehat{\sigma}_i^2/2]$.

The posterior distributions of $\delta_{i,0}$ and $\sigma_i^2$ can be replaced by following Eq (4), and then the equitailed confidence intervals and HPD intervals are constructed by imposing Algorithm 2.

**Algorithm 2**

```
for k = 1 : M do
```

```
Generate datasets xij, for i = 1, 2, j = 1, 2, ..., ni, from the log-
normal distributions with excess zeros;
Calculate δ̂i,1 and σ̂i²;
for l = 1 : m do
  Generate the posterior densities of δi,0 | xij,
  • left-invariant Jeffreys prior: δi,0|xij ~ Beta (ni,0 + 1/2, ni,1
    + 1/2),
  • Jeffreys rule prior: δi,0|xij ~ Beta (ni,0 + 1/2, ni,1 + 3/2),
  • uniform prior: δi,0|xij ~ Beta (ni,0 + 1, ni,1 + 1);
  Generate the posterior densities of σi² | xij,
  • left-invariant Jeffreys prior: σi² | xij ~ IG[(ni,1 − 1)/2, (ni,1 − 1)σ̂i²/2],
  • Jeffreys rule prior: σi² | xij ~ IG(ni,1/2, ni,1σ̂i²/2),
  • uniform prior: σi² | xij ~ IG[(ni,1 − 2)/2, (ni,1 − 2)σ̂i²/2];
  Calculate φ from Eq (4) for the left-invariant Jeffreys, the
  Jeffreys rule, and the uniform priors;
end for
Construct HPD intervals and equitailed confidence intervals for φ
based on the left-invariant Jeffreys, the Jeffreys rule, and the
uniform priors
end for
```

## The Wald log-likelihood method

According to Eq (10), the log-likelihood function is

$$\ln L \propto \quad n_{1,0} \ln \delta_{1,0} + n_{2,0} \ln \delta_{2,0} + n_{1,1} \ln (1 - \delta_{1,0}) + n_{2,1} \ln (1 - \delta_{2,0})$$

$$-\frac{1}{2}\left\{\left[ n_{1,1} \ln \sigma_1^2 + \frac{1}{\sigma_1^2} \sum_{j=1}^{n_{1,1}} (\ln x_{1j} - \mu_1)^2 \right] \right. \tag{17}$$

$$\left. + \left[ n_{2,1} \ln \sigma_2^2 + \frac{1}{\sigma_2^2} \sum_{j=1}^{n_{2,1}} (\ln x_{2j} - \mu_2)^2 \right] \right\}.$$

From Eq (4), the parameter of interest is $\phi$. Subsequently, the log-likelihood function is reparameterized in terms of $\phi$ by substituting $\eta_1 = \phi\eta_2$, $\sigma_1^2 = \ln (1 - \delta_{1,0}) + \ln (\phi^2\eta_2^2 + 1)$, and $\sigma_2^2 = \ln (1 - \delta_{2,0}) + \ln (\eta_2^2 + 1)$ into Eq (17) as follows

$$\ln L \propto \quad n_{1,0} \ln \delta_{1,0} + n_{2,0} \ln \delta_{2,0} + n_{1,1} \ln (1 - \delta_{1,0}) + n_{2,1} \ln (1 - \delta_{2,0})$$

$$-\frac{1}{2}\left\{\left[ n_{1,1} \ln (\ln (1 - \delta_{1,0}) + \ln (\phi^2\eta_2^2 + 1)) + \frac{\sum_{j=1}^{n_{1,1}} (\ln x_{1j} - \mu_1)^2}{\ln (1 - \delta_{1,0}) + \ln (\phi^2\eta_2^2 + 1)} \right] \right. \tag{18}$$

$$\left. + \left[ n_{2,1} \ln (\ln (1 - \delta_{2,0}) + \ln (\eta_2^2 + 1)) + \frac{\sum_{j=1}^{n_{2,1}} (\ln x_{2j} - \mu_2)^2}{\ln (1 - \delta_{2,0}) + \ln (\eta_2^2 + 1)} \right] \right\}.$$

**Theorem 1**. *Let $X_{ij} \sim \Delta(\delta_{i,0}, \mu_i, \sigma_i^2)$, where $i = 1, 2, j = 1, 2, \ldots, n_i$. Let $\ln X_{ij} \sim N(\mu_i, \sigma_i^2)$, for $X_{ij} > 0$. Likewise, let $\phi = \{[( \exp (\sigma_1^2) - \delta_{1,0})/\delta_{1,0}]/[( \exp (\sigma_2^2) - \delta_{2,0})/\delta_{2,0}]\}^{1/2}$, where $\delta_{i,1} = 1 - \delta_{i,0}$ for $i = 1, 2$, be the ratio of CVs of lognormal distributions with excess zeros. The unrestricted MLEs of $\delta_{i,1}, \mu_i,$ and $\sigma_i^2$ are $\hat{\delta}_{i,1} = n_{i,1}/n_i, \hat{\mu}_i = \sum_{j=1}^{n_{i,1}} \ln x_{ij}/n_{i,1},$ and $\hat{\sigma}_i^2 = \sum_{j=1}^{n_{i,1}} (\ln x_{ij} - \hat{\mu}_i)^2/n_i,$ respectively. Consequently, $\hat{\eta}_i = [( \exp (\hat{\sigma}_i^2) - \hat{\delta}_{i,1})/\hat{\delta}_{i,1}]^{1/2}$ and $\hat{\phi} = \{[( \exp (\hat{\sigma}_1^2) - \hat{\delta}_{1,1})/\hat{\delta}_{1,1}]/$*

$[(\exp(\widehat{\sigma}_2^2) - \widehat{\delta}_{2,1})/\widehat{\delta}_{2,1}]\}^{1/2}$. *Therefore, the asymptotic variance of $\widehat{\phi}$ is*

$$Var\left(\widehat{\phi}\right) = \frac{n_{1,1}\phi^4[\sigma_2^2(\eta_2^2+1)]^2 + n_{2,1}[\sigma_1^2(\eta_1^2+1)]^2}{2n_{1,1}n_{2,1}\eta_1^2\eta_2^2}. \tag{19}$$

*Proof.* Following Nam and Kwon [16], since the MLE of $\sigma_i^2$ is $\widehat{\sigma}_i^2 = \sum_{j=1}^{n_{i,1}} (\ln x_{ij} - \widehat{\mu}_i)^2/n_i$, where $\widehat{\mu}_i = \sum_{j=1}^{n_{i,1}} \ln x_{ij}/n_{i,1}$ for $i = 1, 2$, then the log-likelihood function for reparameterization from Eq (17) can be written as

$$\ln L \propto \quad n_{1,0}\ln\delta_{1,0} + n_{2,0}\ln\delta_{2,0} + n_{1,1}\ln(1-\delta_{1,0}) + n_{2,1}\ln(1-\delta_{2,0})$$

$$-\frac{1}{2}\left\{\left[n_{1,1}\ln\left(\ln(1-\delta_{1,0})+\ln(\phi^2\eta_2^2+1)\right) + \frac{n_{1,1}\widehat{\sigma}_1^2}{\ln(1-\delta_{1,0})+\ln(\phi^2\eta_2^2+1)}\right]\right.$$

$$\left.+\left[n_{2,1}\ln\left(\ln(1-\delta_{2,0})+\ln(\eta_2^2+1)\right) + \frac{n_{2,1}\widehat{\sigma}_2^2}{\ln(1-\delta_{2,0})+\ln(\eta_2^2+1)}\right]\right\}.$$

The asymptotic variance of $\widehat{\phi}$ is obtained using the Fisher information which is also written as

$$I(\theta) = -E\left(\frac{\partial^2 \ln L}{\partial\theta^2}\right).$$

By the second-order partial derivative, the Fisher information elements are

$$I_{11} = -E\left(\frac{\partial^2 \ln L}{\partial\phi^2}\right) = \frac{2n_{1,1}\eta_1^2\eta_2^2}{[\sigma_1^2(\eta_1^2+1)]^2}$$

$$I_{22} = -E\left(\frac{\partial^2 \ln L}{\partial\eta_2^2}\right) = \frac{2n_{1,1}\phi^2\eta_1^2}{[\sigma_1^2(\eta_1^2+1)]^2} + \frac{2n_{2,1}\eta_2^2}{[\sigma_2^2(\eta_2^2+1)]^2}$$

$$I_{33} = -E\left(\frac{\partial^2 \ln L}{\partial\delta_{1,0}^2}\right) = \frac{n_1}{\delta_{1,0}(1-\delta_{1,0})}$$

$$I_{44} = -E\left(\frac{\partial^2 \ln L}{\partial\delta_{2,0}^2}\right) = \frac{n_2}{\delta_{2,0}(1-\delta_{2,0})}$$

$$I_{55} = -E\left(\frac{\partial^2 \ln L}{\partial\mu_1^2}\right) = \frac{n_{1,1}}{\sigma_1^2}$$

$$I_{66} = -E\left(\frac{\partial^2 \ln L}{\partial\mu_2^2}\right) = \frac{n_{2,1}}{\sigma_2^2}$$

$$I_{12} = I_{21} = -E\left(\frac{\partial^2 \ln L}{\partial\phi\partial\eta_2}\right) = \frac{2n_{1,1}\eta_1^3}{[\sigma_1^2(\eta_1^2+1)]^2}$$

and the other elements are zeros. By the left-hand block of the matrix $I_n^{-1}(\theta)$, $I^{11}$, the

asymptotic variance of $\widehat{\phi}$ is

$$
\begin{aligned}
Var(\widehat{\phi}) &= I^{11} = \left(I_{11} - \frac{I_{12}^2}{I_{21}}\right)^{-1} \\
&= \frac{n_{1,1}\phi^4[\sigma_2^2(\eta_2^2 + 1)]^2 + n_{2,1}[\sigma_1^2(\eta_1^2 + 1)]^2}{2n_{1,1}n_{2,1}\eta_1^2\eta_2^2},
\end{aligned}
$$

where $\eta_1 = \phi\eta_2$ and $\sigma_i^2 = \ln \delta_{i,1} + \ln (\eta_i^2 + 1)$ for $i = 1, 2$.

Following the MLEs of the parameters $\delta_{i,1} = \widehat{\delta}_{i,1}$, $\eta_i = \widehat{\eta}_i$, $\sigma_i^2 = \widehat{\sigma}_i^2$, for $i = 1, 2$, and $\phi = \widehat{\phi}$, the variance estimate for $\widehat{\phi}$ is

$$
\widehat{Var}\left(\widehat{\phi}\right) = \frac{n_{1,1}\widehat{\phi}^4[\widehat{\sigma}_2^2(\widehat{\eta}_2^2 + 1)]^2 + n_{2,1}[\widehat{\sigma}_1^2(\widehat{\eta}_1^2 + 1)]^2}{2n_{1,1}n_{2,1}\widehat{\eta}_1^2\widehat{\eta}_2^2}. \tag{20}
$$

The asymptotically standard normal distribution is

$$
Z_1 = \frac{\widehat{\phi} - \phi}{\sqrt{\widehat{Var}(\widehat{\phi})}} \sim N(0, 1). \tag{21}
$$

Therefore, the $100(1 - \alpha)\%$ two-sided confidence interval for $\phi$ based on the Wald log-likelihood method is

$$
\phi = \widehat{\phi} \pm z_{1-\alpha/2}\sqrt{\widehat{Var}(\widehat{\phi})}, \tag{22}
$$

where $z_{1-\alpha/2}$ is the $(1 - \alpha/2)$-th percentile of the standard normal distribution.

## The Fieller log-likelihood method

Following Eq (17) and since $\sigma_i^2 = \ln (1 - \delta_{i,0}) + \ln (\eta_i^2 + 1)$, the log-likelihood function can be written as

$$
\begin{aligned}
\ln L \propto\ & n_{1,0} \ln \delta_{1,0} + n_{2,0} \ln \delta_{2,0} + n_{1,1} \ln (1 - \delta_{1,0}) + n_{2,1} \ln (1 - \delta_{2,0}) \\
& - \frac{1}{2}\Bigg\{ \left[n_{1,1} \ln \left(\ln (1 - \delta_{1,0}) + \ln (\eta_1^2 + 1)\right) + \frac{\sum_{j=1}^{n_{1,1}} (\ln x_{1j} - \mu_1)^2}{\ln (1 - \delta_{1,0}) + \ln (\eta_1^2 + 1)}\right] \\
& + \left[n_{2,1} \ln \left(\ln (1 - \delta_{2,0}) + \ln (\eta_2^2 + 1)\right) + \frac{\sum_{j=1}^{n_{2,1}} (\ln x_{2j} - \mu_2)^2}{\ln (1 - \delta_{2,0}) + \ln (\eta_2^2 + 1)}\right]\Bigg\}.
\end{aligned} \tag{23}
$$

**Theorem 2.** *Let $X_{ij} \sim \Delta(\delta_{i,0}, \mu_i, \sigma_i^2)$, where $i = 1, 2, j = 1, 2, \ldots, n_i$, and let $\ln X_{ij} \sim N(\mu_i, \sigma_i^2)$, for $X_{ij} > 0$. The CV of a lognormal distribution with excess zeros is $\eta_i = \{[\exp (\sigma_i^2) - \delta_{i,1}]/\delta_{i,1}\}^{1/2}$, where $\delta_{i,1} = 1 - \delta_{i,0}$ for $i = 1, 2$. Let $\widehat{\delta}_{i,1} = n_{i,1}/n_i$, $\widehat{\mu}_i = \sum_{j=1}^{n_{i,1}} \ln x_{ij}/n_{i,1}$, and $\widehat{\sigma}_i^2 = \sum_{j=1}^{n_{i,1}} (\ln x_{ij} - \widehat{\mu}_i)^2/n_i$ be the unrestricted MLEs of $\delta_{i,1}, \mu_i$, and $\sigma_i^2$, respectively. Likewise, $\widehat{\eta}_i = \{[\exp (\widehat{\sigma}_i^2) - \widehat{\delta}_{i,1}]/\widehat{\delta}_{i,1}\}^{1/2}$. Therefore, the asymptotic variance of*

$\widehat{\eta}_i$ are

$$Var(\widehat{\eta}_i) = \frac{\{[\ln\ (1 - \delta_{i,0}) + \ln\ (\eta_i^2 + 1)](\eta_i^2 + 1)\}^2}{2n_{i,1}\eta_i^2},$$ (24)

*for i* = 1, 2.

*Proof.* The MLEs of the parameters are obtained from the first-order derivative of Eq (23) as $\widehat{\delta}_{i,1} = n_{i,1}/n_i, \widehat{\mu}_i = \sum_{j=1}^{n_{i,1}} \ln\ x_{ij}/n_{i,1}$, and $\widehat{\sigma}_i^2 = \sum_{j=1}^{n_{i,1}} (\ln\ x_{ij} - \widehat{\mu}_i)^2/n_i$. Thus, $\widehat{\eta}_i = \{[\exp\ (\widehat{\sigma}_i^2) - \widehat{\delta}_{i,1}]/\widehat{\delta}_{i,1}\}^{1/2}$, for *i* = 1, 2. Similarly to Theorem 1, the elements of the Fisher information matrix are

$$I_{mn} = -E\left(\frac{\partial^2 \ln\ L}{\partial\eta_m\partial\eta_n}\right)$$

$$= \frac{2n_{i,1}\eta_i^2}{\{[\ln\ (1 - \delta_{i,0}) + \ln\ (\eta_i^2 + 1)](\eta_i^2 + 1)\}^2}$$

for *m* = *n* = 1, 2, *i* = 1, 2, $I_{mn}$ for *m* = *n* = 3, 4, 5, 6 follows from Theorem 1 when $\sigma_i^2 = \ln\ (1 - \delta_{i,0}) + \ln\ (\eta_i^2 + 1)$ and $I_{mn} = 0$ for *m*, *n* = 1, 2, ..., 6 and $m \neq n$. The asymptotic variances of $\widehat{\eta}_1$ and $\widehat{\eta}_2$ are

$$Var(\widehat{\eta}_1) = I^{11} = I_{11}^{-1} = \frac{\{[\ln\ (1 - \delta_{1,0}) + \ln\ (\eta_1^2 + 1)](\eta_1^2 + 1)\}^2}{2n_{1,1}\eta_1^2}$$

and

$$Var(\widehat{\eta}_2) = I^{22} = I_{22}^{-1} = \frac{\{[\ln\ (1 - \delta_{2,0}) + \ln\ (\eta_2^2 + 1)](\eta_2^2 + 1)\}^2}{2n_{2,1}\eta_2^2}.$$

Since $\delta_{i,1} = 1 - \delta_{i,0}$ and the MLEs of $\delta_{i,1}$ and $\sigma_i^2$ are $\widehat{\delta}_{i,1}$ and $\widehat{\sigma}_i^2$, respectively, then the estimated variance of $\widehat{\eta}_i$ is

$$\widehat{Var}(\widehat{\eta}_i) = \frac{[\widehat{\sigma}_i^2(\widehat{\eta}_i^2 + 1)]^2}{2n_{i,1}\widehat{\eta}_i^2},$$ (25)

where $\widehat{\sigma}_i^2 = \ln\ \widehat{\delta}_{i,1} + \ln\ (\widehat{\eta}_i^2 + 1)$, for *i* = 1, 2. According to Fieller [45], the statistic

$$Z_2 = \frac{\widehat{\eta}_1 - \phi\widehat{\eta}_2}{\sqrt{\widehat{Var}(\widehat{\eta}_1) + \phi^2\widehat{Var}(\widehat{\eta}_2)}} \sim N(0, 1).$$ (26)

Therefore, the 100(1 − α)% two-sided confidence interval for φ based on the Fieller log-likelihood method is

$$\phi = \frac{\widehat{\eta}_1\widehat{\eta}_2 \pm \sqrt{(\widehat{\eta}_1\widehat{\eta}_2)^2 - [\widehat{\eta}_1^2 - z_{1-\alpha/2}^2 \widehat{Var}(\widehat{\eta}_1)][\widehat{\eta}_2^2 - z_{1-\alpha/2}^2 \widehat{Var}(\widehat{\eta}_2)]}}{\widehat{\eta}_2^2 - z_{1-\alpha/2}^2 \widehat{Var}(\widehat{\eta}_2)}.$$ (27)

where $z_{1-\alpha/2}$ is the (1 − α/2)-th percentile of a standard normal distribution.

Note that: The variance of the estimator for ratio of CVs from Theorem 1 and the variance of the estimator for CV from Theorem 2 are equal to the variances which are reported by Nam and Kwon [16] when $\delta_{1,1}, \delta_{2,1} = 1$.

**Table 1. The coverage probabilities and average lengths of 95% two-sided confidence intervals for the ratio of CVs of lognormal distributions with excess zeros.**

| $n_1$: $n_2$ | $\delta_1$: $\delta_2$ | $\sigma_1^2 : \sigma_2^2$ | Coverage Probabilities (Average Lengths) | | | | | | | | |
|---|---|---|---|---|---|---|---|---|---|---|---|
| | | | FGCI | Equitailed | | | HPD | | | Wald | Fieller |
| | | | | B-LIJ | B-JR | B-U | B-LIJ | B-JR | B-U | | |
| 25:25 | 0.5:0.5 | 0.5:0.5 | **0.9654** | **0.9835** | **0.9799** | **0.9859** | **0.9872** | **0.9844** | **0.9893** | 0.9061 | **0.9516** |
| | | | (1.3234) | (1.4643) | (1.3937) | (1.5297) | (1.3608) | (1.3037) | (1.4109) | (0.7803) | **(0.8304)** |
| | | 0.5:1.0 | **0.9586** | **0.9719** | **0.9671** | **0.9760** | **0.9725** | **0.9679** | **0.9760** | 0.9330 | 0.9475 |
| | | | (1.1076) | (1.1805) | (1.1220) | (1.2369) | (1.1129) | **(1.0636)** | (1.1579) | (0.7528) | (0.9368) |
| | | 1.0:2.0 | **0.9535** | **0.9573** | 0.9489 | **0.9640** | **0.9555** | 0.9496 | **0.9628** | 0.9407 | 0.8281 |
| | | | (1.7466) | (1.7645) | (1.6379) | (1.9158) | **(1.4734)** | (1.3975) | (1.5569) | (1.0605) | (1.1056) |
| | | 2.0:2.0 | **0.9514** | **0.9559** | 0.9477 | **0.9636** | **0.9573** | **0.9514** | **0.9627** | 0.9127 | 0.8597 |
| | | | (10.9684) | (11.3842) | (9.5735) | (14.3922) | (6.2440) | **(5.5766)** | (7.1173) | (2.5565) | (9.6299) |
| | 0.8:0.8 | 0.5:0.5 | **0.9561** | **0.9675** | **0.9631** | **0.9717** | **0.9711** | **0.9667** | **0.9754** | 0.9215 | **0.9579** |
| | | | (1.1465) | (1.2155) | (1.1818) | (1.2381) | (1.1582) | (1.1287) | (1.1775) | (0.8839) | (0.9510) |
| | | 0.5:1.0 | **0.9555** | **0.9618** | **0.9566** | **0.9665** | **0.9623** | **0.9580** | **0.9679** | 0.9319 | **0.9654** |
| | | | (0.8710) | (0.9016) | (0.8747) | (0.9254) | (0.8672) | (0.8431) | (0.8887) | (0.7100) | (0.8447) |
| | | 1.0:2.0 | **0.9538** | **0.9529** | 0.9477 | **0.9584** | 0.9495 | 0.9455 | **0.9545** | 0.9360 | 0.9438 |
| | | | (1.1656) | (1.1758) | (1.1340) | (1.2198) | (1.0753) | (1.0432) | (1.1082) | (0.8974) | (1.8105) |
| | | 2.0:2.0 | **0.9501** | **0.9503** | 0.9445 | **0.9554** | **0.9511** | 0.9463 | **0.9552** | 0.9147 | **0.9723** |
| | | | (4.1302) | (4.2305) | (3.9862) | (4.5007) | (3.2591) | (3.1220) | (3.4102) | (2.1424) | (3.9999) |
| 25:50 | 0.5:0.5 | 0.5:0.5 | **0.9631** | **0.9804** | **0.9769** | **0.9821** | **0.9776** | **0.9725** | **0.9805** | 0.8478 | 0.8945 |
| | | | (1.1354) | (1.2463) | (1.1768) | (1.3098) | (1.1396) | **(1.0852)** | (1.1826) | (0.6463) | (0.6637) |
| | | 0.5:1.0 | **0.9561** | **0.9735** | **0.9692** | **0.9765** | **0.9714** | **0.9653** | **0.9746** | 0.8943 | **0.9739** |
| | | | (0.9068) | (0.9743) | (0.9210) | (1.0214) | (0.9043) | (0.8615) | (0.9387) | (0.5858) | **(0.6421)** |
| | | 1.0:2.0 | **0.9513** | **0.9600** | **0.9535** | **0.9651** | **0.9583** | **0.9517** | **0.9632** | 0.9133 | **0.9937** |
| | | | (1.4721) | (1.5075) | (1.3869) | (1.6444) | (1.2475) | (1.1733) | (1.3248) | (0.8232) | **(1.1562)** |
| | | 2.0:2.0 | **0.9548** | **0.9577** | 0.9499 | **0.9635** | **0.9543** | 0.9473 | **0.9605** | 0.8630 | **0.9895** |
| | | | (10.3410) | (10.0671) | (8.3466) | (12.8534) | (5.5641) | (4.9297) | (6.4182) | (2.1428) | **(2.7247)** |
| | 0.8:0.8 | 0.5:0.5 | **0.9553** | **0.9683** | **0.9653** | **0.9718** | **0.9706** | **0.9674** | **0.9773** | 0.9011 | 0.9323 |
| | | | (0.9915) | (1.0529) | (1.0218) | (1.0811) | (0.9942) | **(0.9679)** | (1.0177) | (0.7378) | (0.7615) |
| | | 0.5:1.0 | **0.9549** | **0.9660** | **0.9621** | **0.9697** | **0.9650** | **0.9602** | **0.9719** | 0.9099 | **0.9621** |
| | | | (0.7237) | (0.7571) | (0.7351) | (0.7791) | (0.7208) | (0.7016) | (0.7395) | (0.5652) | **(0.6086)** |
| | | 1.0:2.0 | **0.9503** | **0.9548** | **0.9513** | **0.9586** | **0.9581** | **0.9543** | **0.9619** | 0.9227 | **0.9943** |
| | | | (0.9815) | (0.9901) | (0.9555) | (1.0304) | (0.8970) | (0.8709) | (0.9267) | (0.7072) | **(0.8692)** |
| | | 2.0:2.0 | **0.9500** | **0.9530** | 0.9481 | **0.9577** | **0.9581** | **0.9534** | **0.9623** | 0.8895 | **0.9829** |
| | | | (3.7856) | (3.7697) | (3.5480) | (4.0469) | (2.8749) | (2.7497) | (3.0279) | (1.7976) | **(2.1417)** |
| 25:100 | 0.5:0.5 | 0.5:0.5 | **0.9592** | **0.9803** | **0.9756** | **0.9830** | **0.9736** | **0.9659** | **0.9778** | 0.7939 | 0.8203 |
| | | | (1.0311) | (1.1495) | (1.0758) | (1.2122) | (1.0394) | (0.9820) | (1.0809) | (0.5785) | (0.5853) |
| | | 0.5:1.0 | **0.9543** | **0.9745** | **0.9689** | **0.9788** | **0.9685** | **0.9582** | **0.9719** | 0.8447 | 0.9081 |
| | | | (0.7945) | (0.8644) | (0.8109) | (0.9090) | (0.7896) | (0.7474) | (0.8201) | (0.4895) | (0.5095) |
| | | 1.0:2.0 | **0.9526** | **0.9637** | **0.9580** | **0.9684** | **0.9611** | **0.9515** | **0.9649** | 0.8653 | **0.9762** |
| | | | (1.3343) | (1.3555) | (1.2358) | (1.4897) | (1.0972) | (1.0246) | (1.1716) | (0.6752) | (0.7689) |
| | | 2.0:2.0 | **0.9516** | **0.9553** | 0.9462 | **0.9624** | 0.9495 | 0.9401 | **0.9568** | 0.8298 | 0.9202 |
| | | | (9.5088) | (10.2704) | (8.2102) | (13.2450) | (5.3998) | (4.7107) | (6.2762) | (1.9127) | (2.1279) |
| | 0.8:0.8 | 0.5:0.5 | **0.9548** | **0.9670** | **0.9643** | **0.9713** | **0.9675** | **0.9619** | **0.9740** | 0.8671 | 0.8899 |
| | | | (0.9126) | (0.9673) | (0.9359) | (0.9988) | (0.9077) | (0.8810) | (0.9333) | (0.6615) | (0.6709) |
| | | 0.5:1.0 | **0.9554** | **0.9627** | **0.9597** | **0.9654** | **0.9625** | **0.9575** | **0.9697** | 0.8796 | 0.9219 |
| | | | (0.6407) | (0.6715) | (0.6502) | (0.6935) | (0.6332) | (0.6151) | (0.6517) | (0.4826) | (0.4985) |
| | | 1.0:2.0 | **0.9507** | **0.9555** | **0.9520** | **0.9595** | **0.9584** | **0.9555** | **0.9647** | 0.8897 | **0.9595** |
| | | | (0.8574) | (0.8692) | (0.8360) | (0.9081) | (0.7766) | (0.7520) | (0.8048) | (0.5893) | (0.6416) |
| | | 2.0:2.0 | **0.9525** | **0.9512** | 0.9471 | **0.9551** | **0.9550** | **0.9503** | **0.9611** | 0.8657 | 0.9255 |
| | | | (3.5906) | (3.5590) | (3.3357) | (3.8322) | (2.6723) | (2.5466) | (2.8233) | (1.6200) | (1.7423) |

(*Continued*)

**Table 1.** (Continued)

| $n_1 : n_2$ | $\delta_1 : \delta_2$ | $\sigma_1^2 : \sigma_2^2$ | Coverage Probabilities (Average Lengths) | | | | | | | | |
|---|---|---|---|---|---|---|---|---|---|---|---|
| | | | FGCI | Equitailed | | | HPD | | | Wald | Fieller |
| | | | | B-LIJ | B-JR | B-U | B-LIJ | B-JR | B-U | | |
| 50:50 | 0.2:0.2 | 0.5:0.5 | **0.9675** | 0.9885 | 0.9869 | 0.9902 | 0.9904 | 0.9889 | 0.9913 | 0.8934 | 0.9381 |
| | | | (1.4029) | (1.5611) | (1.4427) | (1.6879) | (1.4160) | **(1.3269)** | (1.5007) | (0.6473) | (0.6791) |
| | | 0.5:1.0 | **0.9625** | 0.9778 | 0.9743 | 0.9803 | 0.9755 | 0.9729 | 0.9767 | 0.9166 | 0.9160 |
| | | | (1.2525) | (1.3672) | (1.2661) | (1.4770) | (1.2576) | **(1.1810)** | (1.3302) | (0.7180) | (1.0312) |
| | | 1.0:2.0 | **0.9524** | 0.9617 | 0.9527 | 0.9697 | 0.9631 | 0.9555 | 0.9689 | 0.9392 | 0.7470 |
| | | | (2.4409) | (2.4928) | (2.1804) | (2.9603) | (1.8501) | **(1.7018)** | (2.0355) | (1.1185) | (1.7014) |
| | | 2.0:2.0 | **0.9533** | 0.9613 | 0.9512 | 0.9697 | 0.9571 | 0.9500 | 0.9647 | 0.9063 | 0.8193 |
| | | | (40.7663) | (36.0997) | (21.8269) | (65.5736) | (11.6103) | **(9.0939)** | (15.9990) | (2.8913) | (5.2774) |
| | 0.5:0.5 | 0.5:0.5 | **0.9598** | 0.9823 | 0.9807 | 0.9835 | 0.9835 | 0.9827 | 0.9849 | 0.9105 | 0.9327 |
| | | | **(0.7841)** | (0.8854) | (0.8689) | (0.8957) | (0.8595) | (0.8443) | (0.8689) | (0.5544) | (0.5713) |
| | | 0.5:1.0 | **0.9549** | 0.9679 | 0.9648 | 0.9701 | 0.9689 | 0.9671 | 0.9715 | 0.9281 | 0.9475 |
| | | | **(0.6816)** | (0.7365) | (0.7209) | (0.7477) | (0.7210) | (0.7062) | (0.7316) | (0.5435) | (0.6019) |
| | | 1.0:2.0 | **0.9508** | 0.9583 | 0.9545 | 0.9616 | 0.9589 | 0.9546 | 0.9619 | 0.9427 | **0.9540** |
| | | | (0.9225) | (0.9453) | (0.9228) | (0.9676) | (0.8966) | **(0.8774)** | (0.9154) | (0.7578) | (1.0524) |
| | | 2.0:2.0 | **0.9500** | 0.9528 | 0.9483 | **0.9561** | 0.9556 | 0.9521 | 0.9587 | 0.9277 | **0.9967** |
| | | | (2.7610) | (2.8132) | (2.7145) | (2.9196) | (2.3746) | **(2.3092)** | (2.4431) | (1.7547) | (2.8522) |
| | 0.8:0.8 | 0.5:0.5 | **0.9563** | 0.9671 | 0.9643 | 0.9697 | 0.9677 | 0.9658 | 0.9701 | 0.9279 | 0.9477 |
| | | | **(0.7258)** | (0.7740) | (0.7646) | (0.7788) | (0.7565) | (0.7477) | (0.7610) | (0.6165) | (0.6389) |
| | | 0.5:1.0 | **0.9532** | 0.9570 | 0.9547 | 0.9606 | 0.9585 | 0.9565 | 0.9615 | 0.9365 | **0.9583** |
| | | | (0.5606) | (0.5818) | (0.5743) | (0.5881) | (0.5715) | (0.5644) | (0.5777) | (0.4986) | **(0.5417)** |
| | | 1.0:2.0 | **0.9529** | 0.9500 | 0.9467 | **0.9524** | 0.9503 | 0.9482 | 0.9537 | 0.9443 | **0.9637** |
| | | | (0.7013) | (0.7099) | (0.7000) | (0.7197) | **(0.6869)** | (0.6779) | (0.6960) | (0.6267) | (0.7872) |
| | | 2.0:2.0 | **0.9501** | 0.9502 | 0.9473 | **0.9525** | 0.9549 | 0.9532 | 0.9572 | 0.9347 | **0.9973** |
| | | | (1.8696) | (1.8701) | (1.8372) | (1.9029) | (1.7035) | **(1.6779)** | (1.7292) | (1.4471) | (1.7632) |
| 50:100 | 0.2:0.2 | 0.5:0.5 | **0.9658** | 0.9882 | 0.9855 | 0.9891 | 0.9873 | 0.9830 | 0.9879 | 0.8433 | 0.8855 |
| | | | (1.2156) | (1.3690) | (1.2439) | (1.4802) | (1.2081) | **(1.1179)** | (1.2694) | (0.5533) | (0.5645) |
| | | 0.5:1.0 | **0.9630** | 0.9821 | 0.9784 | 0.9839 | 0.9817 | 0.9752 | 0.9824 | 0.9113 | **0.9712** |
| | | | (1.0392) | (1.1399) | (1.0410) | (1.2270) | (1.0282) | (0.9547) | (1.0788) | (0.5707) | **(0.6263)** |
| | | 1.0:2.0 | **0.9549** | 0.9655 | 0.9599 | 0.9703 | 0.9662 | 0.9575 | 0.9695 | 0.9301 | **0.9753** |
| | | | (2.0945) | (2.2044) | (1.8812) | (2.6662) | (1.6118) | (1.4499) | (1.7906) | (0.8727) | **(0.5866)** |
| | | 2.0:2.0 | **0.9538** | 0.9571 | 0.9485 | **0.9642** | 0.9548 | 0.9444 | 0.9610 | 0.8528 | **0.9797** |
| | | | (67.8516) | (61.5960) | (34.1736) | (164.6780) | (13.8966) | (10.3774) | (22.4465) | (2.3299) | **(3.0886)** |
| | 0.5:0.5 | 0.5:0.5 | **0.9635** | 0.9821 | 0.9806 | 0.9829 | 0.9805 | 0.9774 | 0.9819 | 0.8793 | 0.9045 |
| | | | **(0.6718)** | (0.7570) | (0.7399) | (0.7661) | (0.7309) | (0.7154) | (0.7383) | (0.4717) | (0.4779) |
| | | 0.5:1.0 | **0.9559** | 0.9739 | 0.9723 | 0.9752 | 0.9715 | 0.9679 | 0.9737 | 0.9097 | **0.9511** |
| | | | (0.5556) | (0.6052) | (0.5918) | (0.6130) | (0.5882) | (0.5758) | (0.5951) | (0.4297) | **(0.4496)** |
| | | 1.0:2.0 | **0.9505** | 0.9586 | 0.9567 | 0.9611 | 0.9617 | 0.9573 | 0.9643 | 0.9262 | **0.9882** |
| | | | (0.7576) | (0.7811) | (0.7609) | (0.7979) | (0.7341) | (0.7172) | (0.7473) | (0.5904) | **(0.6832)** |
| | | 2.0:2.0 | **0.9503** | 0.9529 | 0.9502 | **0.9563** | 0.9583 | 0.9540 | 0.9609 | 0.9050 | **0.9792** |
| | | | (2.5209) | (2.5261) | (2.4280) | (2.6227) | (2.0993) | (2.0352) | (2.1614) | (1.4996) | **(1.6864)** |

(*Continued*)

**Table 1.** (Continued)

| $n_1 : n_2$ | $\delta_1 : \delta_2$ | $\sigma_1^2 : \sigma_2^2$ | Coverage Probabilities (Average Lengths) | | | | | | | | |
|---|---|---|---|---|---|---|---|---|---|---|---|
| | | | FGCI | Equitailed | | | HPD | | | Wald | Fieller |
| | | | | B-LIJ | B-JR | B-U | B-LIJ | B-JR | B-U | | |
| | 0.8:0.8 | 0.5:0.5 | **0.9535** | **0.9659** | **0.9639** | **0.9668** | **0.9657** | **0.9630** | **0.9684** | 0.9121 | 0.9310 |
| | | | **(0.6263)** | (0.6669) | (0.6586) | (0.6734) | (0.6496) | (0.6418) | (0.6555) | (0.5218) | (0.5300) |
| | | 0.5:1.0 | **0.9529** | **0.9593** | **0.9575** | **0.9616** | **0.9575** | **0.9550** | **0.9609** | 0.9172 | 0.9489 |
| | | | **(0.4624)** | (0.4817) | (0.4756) | (0.4872) | (0.4710) | (0.4654) | (0.4763) | (0.3999) | (0.4148) |
| | | 1.0:2.0 | **0.9531** | 0.9518 | 0.9495 | **0.9543** | **0.9517** | 0.9489 | **0.9546** | 0.9286 | **0.9765** |
| | | | (0.5747) | (0.5844) | (0.5767) | (0.5925) | (0.5615) | (0.5547) | (0.5688) | (0.4967) | **(0.5467)** |
| | | 2.0:2.0 | **0.9529** | 0.9518 | 0.9485 | **0.9531** | **0.9599** | **0.9575** | **0.9618** | 0.9165 | **0.9705** |
| | | | (1.6544) | (1.6684) | (1.6388) | (1.7006) | (1.4996) | (1.4773) | (1.5243) | (1.2334) | **(1.3380)** |
| 100:100 | 0.2:0.2 | 0.5:0.5 | **0.9637** | **0.9885** | **0.9875** | **0.9888** | **0.9908** | **0.9904** | **0.9917** | 0.8920 | 0.9182 |
| | | | (0.7637) | (0.8899) | (0.8673) | (0.9033) | (0.8618) | (0.8411) | (0.8735) | (0.4694) | (0.4801) |
| | | 0.5:1.0 | **0.9616** | **0.9755** | **0.9732** | **0.9781** | **0.9785** | **0.9772** | **0.9804** | 0.9227 | 0.9277 |
| | | | (0.7215) | (0.8006) | (0.7793) | (0.8150) | (0.7834) | (0.7633) | (0.7971) | (0.5276) | (0.5839) |
| | | 1.0:2.0 | **0.9513** | **0.9607** | **0.9556** | **0.9641** | **0.9615** | **0.9579** | **0.9639** | 0.9433 | 0.9315 |
| | | | (1.0450) | (1.0894) | (1.0542) | (1.1220) | (1.0187) | (0.9904) | (1.0444) | (0.8018) | (1.3385) |
| | | 2.0:2.0 | **0.9522** | **0.9589** | **0.9543** | **0.9627** | **0.9599** | **0.9571** | **0.9631** | 0.9270 | **0.9835** |
| | | | (3.4830) | (3.5748) | (3.3877) | (3.7837) | (2.8333) | (2.7227) | (2.9520) | (1.8358) | (2.4826) |
| | 0.5:0.5 | 0.5:0.5 | **0.9562** | **0.9829** | **0.9820** | **0.9836** | **0.9831** | **0.9820** | **0.9835** | 0.9056 | 0.9177 |
| | | | **(0.5149)** | (0.5854) | (0.5805) | (0.5878) | (0.5765) | (0.5719) | (0.5788) | (0.3961) | (0.4021) |
| | | 0.5:1.0 | **0.9543** | **0.9725** | **0.9710** | **0.9739** | **0.9706** | **0.9701** | **0.9723** | 0.9307 | 0.9420 |
| | | | **(0.4555)** | (0.4926) | (0.4879) | (0.4958) | (0.4873) | (0.4826) | (0.4904) | (0.3895) | (0.4093) |
| | | 1.0:2.0 | **0.9513** | **0.9558** | **0.9546** | **0.9588** | **0.9581** | **0.9555** | **0.9603** | 0.9457 | **0.9545** |
| | | | (0.5917) | (0.6083) | (0.6020) | (0.6142) | (0.5957) | **(0.5897)** | (0.6013) | (0.5379) | (0.6305) |
| | | 2.0:2.0 | **0.9523** | **0.9529** | 0.9510 | **0.9550** | **0.9590** | **0.9585** | **0.9622** | 0.9387 | **0.9918** |
| | | | (1.4645) | (1.4979) | (1.4795) | (1.5164) | (1.4009) | (1.3850) | (1.4158) | (1.1982) | **(1.3712)** |
| | 0.8:0.8 | 0.5:0.5 | **0.9520** | **0.9681** | **0.9674** | **0.9699** | **0.9682** | **0.9667** | **0.9688** | 0.9345 | 0.9434 |
| | | | **(0.4892)** | (0.5242) | (0.5212) | (0.5255) | (0.5174) | (0.5146) | (0.5188) | (0.4333) | (0.4410) |
| | | 0.5:1.0 | **0.9507** | **0.9599** | **0.9582** | **0.9603** | **0.9601** | **0.9579** | **0.9611** | 0.9379 | 0.9499 |
| | | | **(0.3813)** | (0.3973) | (0.3948) | (0.3992) | (0.3932) | (0.3908) | (0.3951) | (0.3531) | (0.3677) |
| | | 1.0:2.0 | **0.9512** | 0.9498 | 0.9492 | **0.9518** | **0.9501** | 0.9487 | **0.9522** | 0.9437 | **0.9623** |
| | | | (0.4669) | (0.4715) | (0.4685) | (0.4746) | **(0.4642)** | (0.4612) | (0.4671) | (0.4406) | (0.4900) |
| | | 2.0:2.0 | **0.9510** | **0.9505** | 0.9485 | **0.9524** | **0.9554** | **0.9527** | **0.9563** | 0.9415 | **0.9786** |
| | | | (1.1197) | (1.1283) | (1.1201) | (1.1361) | (1.0821) | **(1.0748)** | (1.0892) | (0.9968) | (1.0932) |

**Algorithm 3**
```
for k = 1 : M do
    Generate datasets xᵢⱼ, for i = 1, 2, j = 1, 2, ..., nᵢ, from the log-
normal distributions with excess zeros;
    Calculate δ̂ᵢ,₁ and σ̂ᵢ²;
    Calculate Var̂(φ̂) from Eq (20) and Var̂(η̂ᵢ) from Eq (25);
    Construct two-sided confidence intervals for φ based on the Wald
log-likelihood and Fieller log-likelihood methods
end for
```

## Results and discussion

### Simulation studies

Simulation studies were conducted to compare the performances of the methods used to construct the confidence intervals using FGCI, the Bayesian methods (the left-invariant Jeffreys, the Jeffreys rule, and the uniform priors), and the Wald and Fieller log-likelihood methods. The optimal method was the one with a coverage probability equal to or greater than the nominal confidence level of 0.95 together with the shortest average length. Following Wu and Hsieh [46], cases that were expected to have non-zero values of less than 10 were not considered. Sample sizes ($n_i$), $\delta_{i,1}$, and $\sigma_i^2$ were set as reported in Table 1. For all of the simulations, 15,000 runs were generated and 5,000 replicates were defined for the FGCI and Bayesian methods via Monte Carlo simulation using RStudio version 1.1.463.

Table 1 and Figs 1–3 present the coverage probabilities and average lengths of the confidence intervals for the various methods. The results show that the coverage probabilities of FGCI were consistently close to the nominal confidence level of 0.95 for all cases. The coverage probabilities of the Bayesian method using the uniform prior (B-U) for both the equitailed confidence interval and HPD interval were greater than or close to the nominal confidence level of 0.95 for all cases. The Bayesian methods using the left-invariant Jeffreys (B-LIJ) and the Jeffreys rule (B-JR) priors based on equitailed confidence intervals and HPD intervals attained coverage probabilities greater than or close to the nominal confidence level of 0.95 in almost every case. However, those attained by the Wald log-likelihood method were less than the nominal confidence level 0.95 for all cases whereas those produced by the Fieller log-likelihood method were greater than or close to the nominal confidence level 0.95 in some cases.

The average lengths of B-JR based on the HPD interval was the shortest for most of the cases when the sample size was small ($n_1$ and/or $n_2 = 25$). For large sample sizes ($n_1$, $n_2 = 50$, 100), B-JR based on the HPD interval had mainly narrow average lengths for the cases of $\delta_{1,1}$, $\delta_{2,1} = 0.2$ for all variances and $\delta_{1,1}$, $\delta_{2,1} = 0.5, 0.8$ together with $\sigma_1^2$, $\sigma_2^2 = 1, 2$, while those of FGCI were the shortest for cases with small variance(s) ($\sigma_1^2$ and/or $\sigma_2^2 = 0.5$).

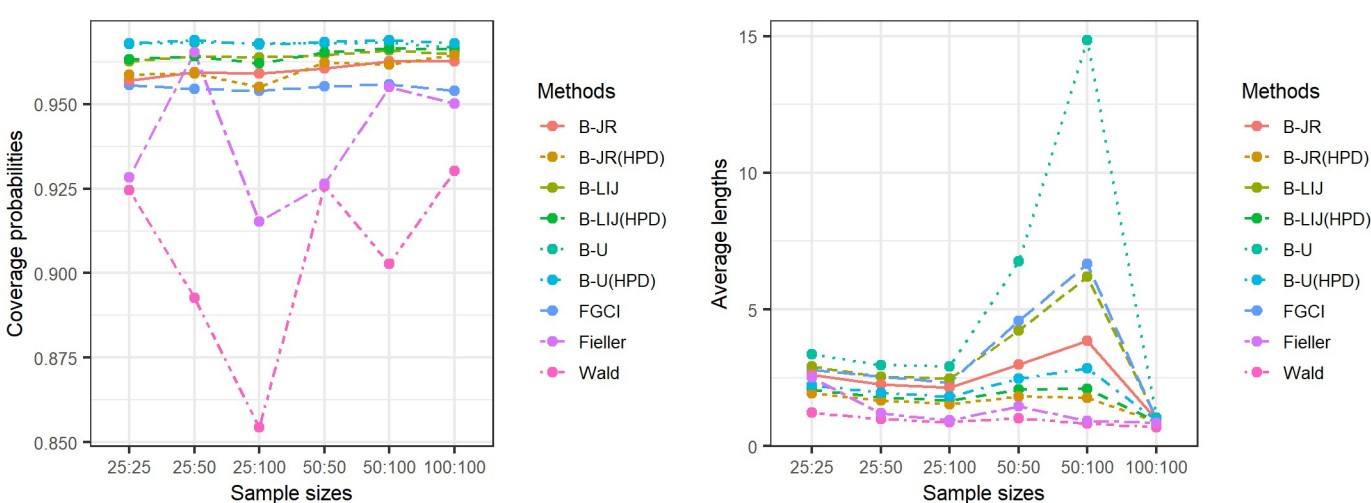

**Fig 1. Line graphs for comparing the coverage probabilities (CP) and average lengths (AL) of all methods in cases of the different sample sizes.**

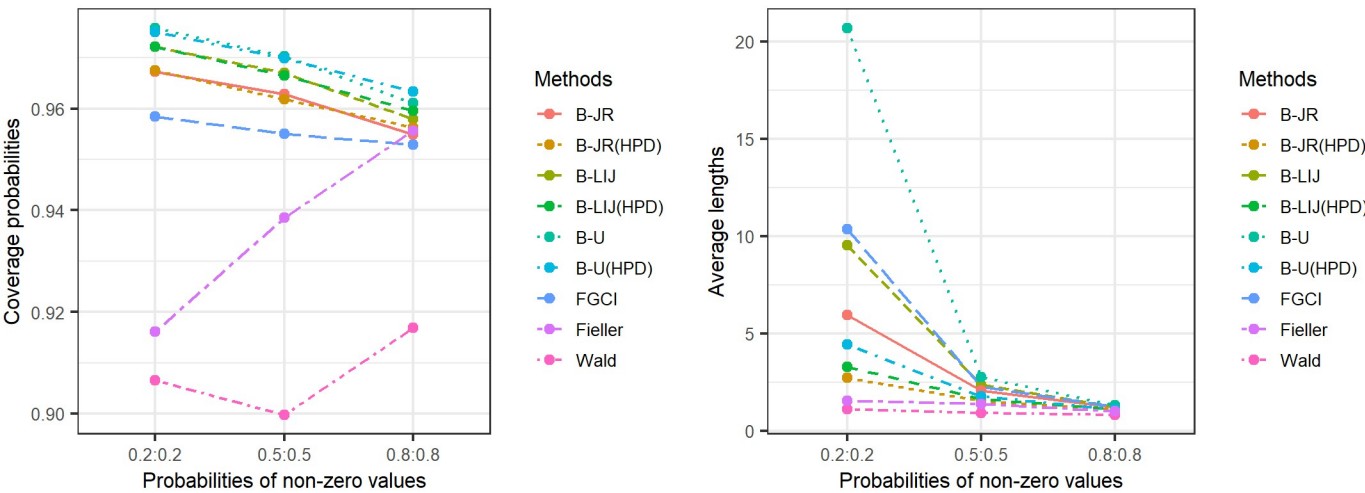

**Fig 2. Line graphs for comparing the coverage probabilities (CP) and average lengths (AL) of all methods in cases of the different probabilities of non-zero values.**

## An empirical example

As previously mentioned, the CV can be used to measure the dispersion in a dataset, especially in cases like rainfall data that conform to a lognormal distribution with excess zeros. Therefore, daily rainfall data from the central and southern regions (Chumphon province) in August 2017, collected by the Central and Southern Region Irrigation Hydrology Center were used to construct confidence intervals for evaluating the proposed methods. The datasets were shown in Tables 2 and 3. The data with zero values implied a binomial distribution whereas the positive observations for both regions were skewed, as shown in Fig 4, and so log transformation of the positive values was applied. It is imperative to check the distribution of the data, and so a minimum Akaike information criteria (AIC) analysis was conducted. The AIC values of the rainfall data from the central and southern regions for normal, lognormal, and Cauchy

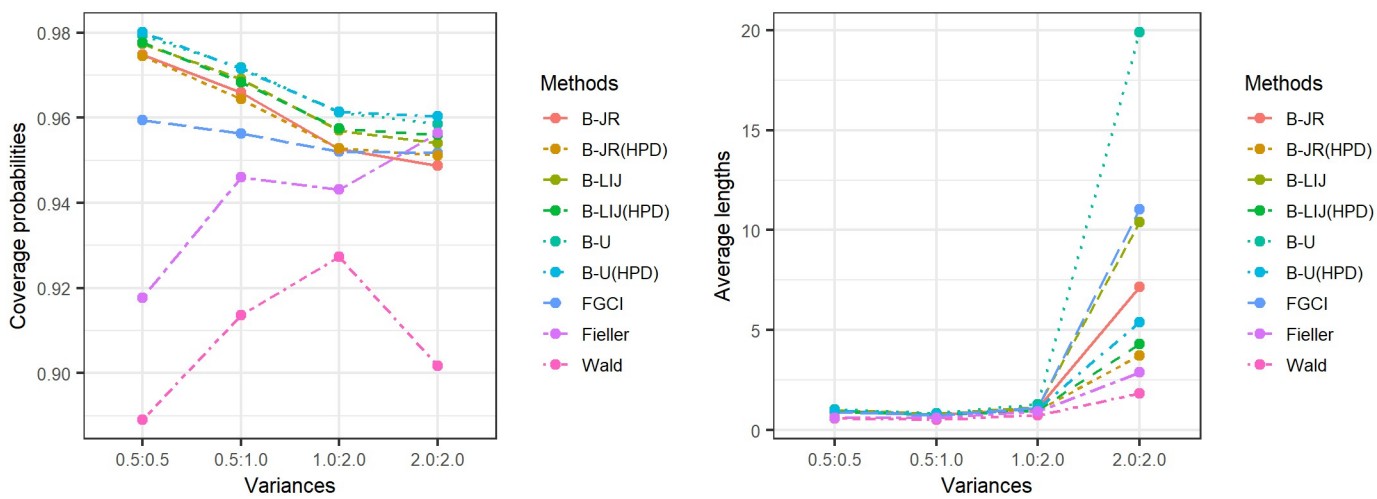

**Fig 3. Line graphs for comparing the coverage probabilities (CP) and average lengths (AL) of all methods in cases of the different variances.**

**Table 2. Daily rainfall data from central region on August, 2017.**

| Date | Stations | | | | | | | | | | | | |
|---|---|---|---|---|---|---|---|---|---|---|---|---|---|
| | N.67 | C.2 | C.13 | C.30 | Ct.4 | Ct.5A | Ct.7 | Ct.9 | Ct.2A | S.9 | S.13 | S.28 | T.7 |
| 1 | 0.0 | 0.0 | 0.0 | 0.0 | 0.0 | 0.0 | 0.0 | 0.0 | 0.0 | 0.0 | 6.4 | 0.0 | 0.0 |
| 2 | 0.0 | 0.0 | 0.0 | 9.2 | 4.2 | 0.0 | 0.0 | 0.0 | 17.0 | 0.0 | 9.6 | 2.2 | 0.0 |
| 3 | 0.0 | 0.0 | 0.0 | 1.1 | 2.6 | 0.0 | 0.0 | 0.0 | 0.0 | 0.0 | 1.5 | 0.0 | 0.0 |
| 4 | 8.2 | 0.0 | 0.0 | 0.0 | 0.6 | 0.0 | 28.5 | 6.1 | 1.4 | 26.2 | 0.0 | 0.0 | 0.0 |
| 5 | 3.9 | 4.4 | 36.1 | 0.0 | 0.4 | 0.0 | 10.0 | 3.0 | 0.0 | 56.3 | 15.2 | 8.6 | 0.0 |
| 6 | 0.0 | 0.0 | 2.2 | 0.0 | 0.0 | 10.5 | 0.0 | 0.0 | 1.8 | 4.1 | 3.2 | 4.5 | 2.9 |
| 7 | 0.0 | 0.0 | 0.4 | 13.5 | 0.0 | 19.2 | 1.0 | 10.2 | 4.1 | 1.5 | 2.5 | 1.3 | 27.9 |
| 8 | 4.0 | 19.1 | 0.0 | 0.0 | 0.0 | 7.8 | 0.0 | 0.0 | 3.1 | 7.8 | 0.0 | 0.0 | 0.0 |
| 9 | 0.0 | 0.0 | 0.0 | 0.0 | 0.0 | 0.0 | 0.0 | 0.0 | 0.0 | 0.0 | 0.0 | 0.0 | 0.0 |
| 10 | 11.0 | 0.0 | 17.4 | 0.0 | 3.0 | 0.0 | 0.0 | 0.0 | 0.0 | 8.0 | 0.0 | 0.0 | 0.0 |
| 11 | 0.0 | 0.0 | 0.0 | 3.8 | 0.0 | 0.0 | 0.0 | 0.0 | 0.0 | 0.0 | 0.0 | 0.0 | 0.0 |
| 12 | 0.0 | 0.0 | 9.5 | 0.0 | 18.8 | 0.0 | 0.0 | 0.0 | 0.0 | 0.0 | 0.0 | 0.0 | 0.0 |
| 13 | 0.0 | 0.0 | 0.0 | 0.0 | 0.0 | 0.0 | 0.0 | 0.0 | 0.0 | 0.0 | 0.0 | 0.0 | 20.0 |
| 14 | 1.3 | 0.9 | 0.0 | 0.0 | 0.0 | 0.0 | 13.5 | 10.8 | 0.0 | 0.0 | 0.0 | 0.0 | 1.1 |
| 15 | 44.2 | 9.3 | 10.8 | 3.4 | 70.0 | 0.0 | 8.3 | 3.0 | 52.0 | 9.3 | 0.0 | 0.0 | 0.0 |
| 16 | 12.0 | 6.4 | 26.6 | 22.2 | 54.5 | 54.8 | 11.5 | 25.4 | 9.2 | 12.3 | 5.7 | 2.2 | 12.4 |
| 17 | 2.6 | 0.0 | 9.5 | 12.5 | 15.3 | 65.6 | 18.5 | 2.0 | 9.4 | 0.0 | 2.6 | 12.8 | 16.2 |
| 18 | 1.0 | 11.8 | 12.3 | 21.6 | 12.9 | 14.3 | 4.0 | 46.7 | 0.8 | 6.2 | 18.8 | 6.8 | 2.2 |
| 19 | 0.0 | 0.6 | 44.8 | 5.1 | 0.0 | 0.0 | 36.5 | 17.0 | 14.7 | 2.1 | 0.0 | 26.7 | 9.5 |
| 20 | 67.8 | 17.9 | 3.8 | 7.6 | 0.2 | 17.2 | 5.5 | 8.0 | 9.9 | 0.0 | 20.5 | 0.5 | 40.0 |
| 21 | 0.0 | 0.8 | 0.0 | 15.4 | 0.2 | 6.4 | 0.0 | 14.2 | 0.0 | 0.0 | 0.0 | 0.0 | 0.0 |
| 22 | 15.0 | 0.0 | 0.0 | 8.5 | 3.1 | 0.0 | 2.2 | 5.4 | 0.0 | 0.0 | 0.0 | 0.0 | 0.0 |
| 23 | 0.0 | 0.0 | 0.0 | 13.9 | 0.0 | 8.3 | 1.5 | 0.0 | 0.0 | 10.0 | 0.0 | 0.0 | 10.5 |
| 24 | 38.5 | 2.3 | 24.8 | 6.5 | 8.0 | 5.4 | 0.0 | 0.8 | 30.0 | 6.2 | 6.4 | 0.0 | 0.0 |
| 25 | 13.0 | 7.6 | 0.0 | 9.3 | 10.0 | 10.2 | 2.0 | 3.4 | 0.0 | 0.0 | 3.3 | 0.0 | 0.0 |
| 26 | 0.0 | 5.1 | 0.0 | 1.9 | 0.3 | 7.4 | 0.0 | 44.7 | 0.0 | 13.6 | 26.8 | 26.3 | 3.9 |
| 27 | 24.7 | 55.7 | 12.7 | 18.2 | 30.0 | 0.0 | 35.0 | 22.1 | 13.4 | 13.0 | 62.5 | 4.6 | 89.9 |
| 28 | 11.0 | 14.7 | 0.0 | 0.0 | 53.6 | 10.2 | 14.0 | 1.0 | 7.0 | 10.6 | 0.0 | 2.5 | 4.9 |
| 29 | 4.8 | 2.2 | 0.0 | 0.0 | 4.3 | 15.4 | 0.0 | 0.0 | 21.1 | 0.0 | 3.1 | 0.0 | 0.0 |
| 30 | 4.4 | 24.8 | 0.0 | 0.0 | 13.6 | 0.0 | 8.0 | 16.5 | 19.1 | 7.3 | 2.4 | 5.2 | 41.9 |
| 31 | 0.0 | 0.0 | 0.0 | 12.7 | 1.3 | 0.0 | 2.0 | 3.8 | 0.0 | 0.0 | 0.0 | 0.0 | 7.2 |

Source: Central region irrigation hydrology center (http://hydro-5.rid.go.th/)

distributions were 1491.7130, 1251.7700, and 1390.5430 and 952.6130, 840.1782, and 919.0321, respectively. Thus, the lognormal distribution was suitable for both datasets. This was further confirmed using normal Q-Q plots (Fig 5). The summary statistics for the rainfall data from the central and southern regions were $n_1 = 390$, $\widehat{\delta}_{1,1} = 0.4667$, $\widehat{\mu}_1 = 1.7573$, $\widehat{\sigma}_1^2 = 1.6636$, $\widehat{\eta}_1 = 0.7340$ and $n_2 = 248$, $\widehat{\delta}_{2,1} = 0.5121$, $\widehat{\mu}_2 = 1.7913$, $\widehat{\sigma}_2^2 = 1.1872$, $\widehat{\eta}_2 = 0.6083$, respectively. The ratio of CVs between two regions was $\phi = 1.2066$, and the 95% confidence intervals for $\phi$ are reported in Table 4.

The lower and upper bounds from the results indicate that the dispersion of rainfall in the central region was more than the southern region. This is because the southern region has abundant precipitation throughout the year due to being located on the peninsula surrounded by the Andaman Sea and the Gulf of Thailand. The central region is located on the plains that

**Table 3. Daily rainfall data from southern region on August, 2017.**

| Date | Stations | | | | | | | |
|---|---|---|---|---|---|---|---|---|
| | 100191 | 100251 | 100261 | 100271 | 100281 | 100291 | 100301 | 100311 |
| 1 | 2.0 | 0.0 | 4.7 | 13.6 | 2.3 | 2.3 | 14.7 | 11.3 |
| 2 | 0.0 | 0.0 | 4.8 | 2.2 | 3.9 | 3.9 | 3.2 | 0.0 |
| 3 | 6.0 | 0.0 | 3.4 | 4.1 | 0.0 | 0.0 | 8.0 | 2.0 |
| 4 | 5.0 | 0.0 | 2.9 | 0.0 | 0.0 | 0.0 | 40.8 | 5.0 |
| 5 | 0.0 | 0.0 | 5.3 | 0.0 | 0.0 | 0.0 | 0.0 | 0.0 |
| 6 | 0.0 | 0.0 | 2.6 | 0.0 | 0.0 | 0.0 | 13.9 | 2.1 |
| 7 | 0.0 | 0.0 | 0.0 | 0.0 | 0.0 | 0.0 | 0.0 | 0.0 |
| 8 | 0.0 | 0.0 | 30.0 | 0.0 | 0.0 | 0.0 | 7.2 | 1.3 |
| 9 | 0.0 | 0.0 | 0.0 | 0.0 | 0.0 | 0.0 | 0.0 | 0.0 |
| 10 | 0.0 | 0.0 | 0.0 | 0.0 | 0.0 | 0.0 | 0.0 | 0.0 |
| 11 | 0.0 | 0.0 | 0.0 | 0.0 | 0.0 | 0.0 | 8.6 | 0.0 |
| 12 | 0.0 | 0.0 | 0.0 | 0.0 | 0.0 | 0.0 | 0.0 | 0.0 |
| 13 | 0.0 | 0.0 | 0.0 | 0.0 | 0.0 | 0.0 | 0.0 | 0.0 |
| 14 | 63.0 | 10.5 | 3.6 | 6.1 | 42.2 | 42.2 | 18.6 | 10.6 |
| 15 | 0.0 | 7.0 | 16.1 | 22.6 | 18.3 | 18.3 | 1.6 | 0.0 |
| 16 | 3.0 | 0.0 | 17.0 | 7.4 | 5.2 | 5.2 | 1.8 | 0.0 |
| 17 | 25.0 | 5.0 | 9.0 | 21.6 | 17.3 | 17.3 | 32.7 | 27.0 |
| 18 | 16.0 | 6.0 | 7.5 | 16.2 | 11.3 | 11.3 | 20.2 | 12.8 |
| 19 | 2.0 | 0.4 | 7.0 | 14.0 | 3.7 | 3.7 | 5.0 | 1.5 |
| 20 | 9.0 | 0.0 | 3.1 | 6.3 | 8.4 | 8.4 | 11.2 | 1.7 |
| 21 | 0.0 | 0.0 | 10.6 | 3.6 | 0.0 | 0.0 | 0.0 | 0.0 |
| 22 | 0.0 | 4.0 | 6.0 | 12.2 | 4.4 | 4.4 | 0.3 | 0.0 |
| 23 | 6.0 | 0.7 | 16.9 | 25.7 | 0.0 | 0.0 | 6.6 | 1.5 |
| 24 | 0.0 | 0.8 | 0.9 | 24.4 | 0.0 | 0.0 | 0.0 | 2.1 |
| 25 | 8.0 | 0.4 | 19.1 | 27.1 | 0.0 | 0.0 | 8.0 | 6.0 |
| 26 | 6.0 | 0.0 | 9.2 | 26.4 | 0.0 | 0.0 | 9.2 | 3.0 |
| 27 | 1.0 | 0.5 | 6.2 | 13.2 | 0.0 | 0.0 | 3.9 | 1.2 |
| 28 | 8.0 | 1.0 | 27.8 | 18.7 | 4.5 | 4.5 | 10.0 | 8.5 |
| 29 | 0.0 | 1.2 | 0.0 | 0.0 | 0.0 | 0.0 | 0.0 | 1.2 |
| 30 | 0.0 | 0.0 | 0.0 | 0.0 | 0.0 | 0.0 | 1.4 | 0.0 |
| 31 | 0.0 | 0.0 | 12.3 | 0.0 | 0.0 | 0.0 | 0.0 | 0.0 |

Source: Southern region irrigation hydrology center (http://hydro-8.com/main/Submenu/3-RAIN/3-RAIN-02.html)

cause irregular precipitation, thus the dispersion of the rainfall data is larger than in the southern region.

## Conclusion

FGCI, Bayesian methods based on the left-invariant Jeffreys, Jeffreys rule, and uniform priors, and the Wald and Fieller log-likelihood methods were used to construct the confidence intervals for the ratio of CVs of lognormal distributions with excess zeros. Coverage probabilities and the average lengths were used to evaluate the performance of the proposed methods.

The simulation results indicate that the coverage probabilities for all cases of the FGCI and Bayesian methods using the uniform prior and almost all cases of the Bayesian method using the left-invariant Jeffreys and Jeffreys rule priors were close to or greater than the target.

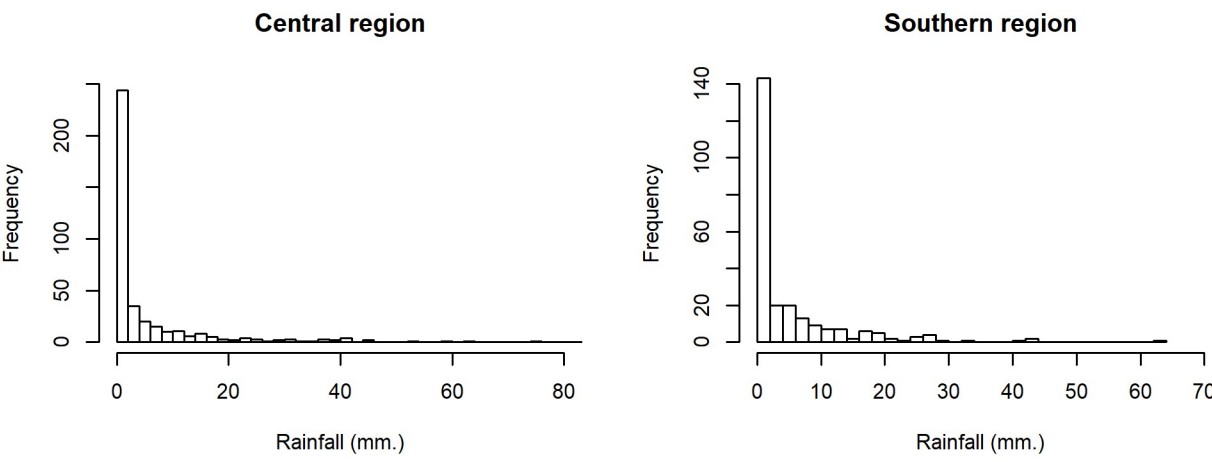

**Fig 4. Density of rainfall data sets from central and southern regions in Thailand.**

However, when considering the average lengths, the Bayesian method using the Jeffreys rule prior based on the HPD interval produced the shortest ones in cases of small sample sizes and a large sample size together with a small expected number of non-zero observations and a large variance, while FGCI was optimal for the other cases. Therefore, the HPD Bayesian method using the Jeffreys rule prior and the FGCI method are suitable for constructing confidence intervals for the ratio of CVs of lognormal distributions with excess zeros.

Nam and Kwon [16] introduced the Wald-type and Fieller-type methods for the ratio of CVs of lognormal distributions that were appropriate for medium sample sizes. In the present study, this method was extended for a lognormal distribution with zero-inflated observations. However, the coverage probabilities of the Wald log-likelihood method were less than the target for all cases whereas those of the Fieller log-likelihood method were greater than the target for a few cases when the probability of non-zero values was more than half and for a large variance. Moreover, the average lengths of these methods were wider than the FGCI and Bayesian methods. Hence, the Wald and Fieller log-likelihood methods are not recommended for

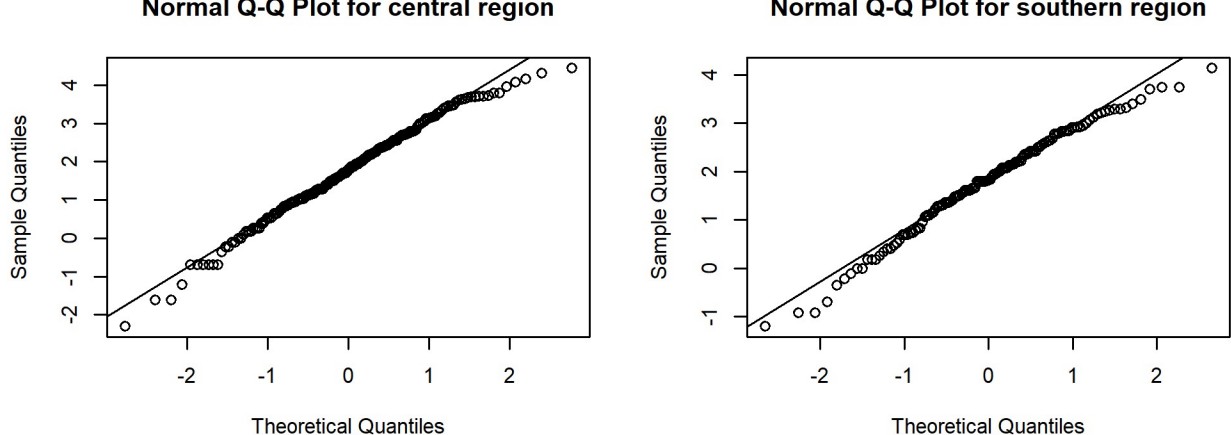

**Fig 5. Normal Q-Q plot of log-transformed for rainfall data sets from central and southern regions in Thailand.**

**Table 4. The 95% two-sided confidence intervals for the ratio of CVs of daily rainfall data between central and southern region in August, 2017.**

| Methods | Confidence intervals | | |
|---|---|---|---|
| | Lower | Upper | Length |
| FGCI | 1.0540 | 1.8070 | 0.7530 |
| B-LIJ (Equitailed) | 1.0416 | 1.8182 | 0.7766 |
| B-JR (Equitailed) | 1.0514 | 1.7957 | 0.7443 |
| B-U (Equitailed) | 1.0420 | 1.8296 | 0.7876 |
| B-LIJ (HPD) | 0.9905 | 1.7545 | 0.7640 |
| B-JR (HPD) | 1.0119 | 1.7532 | 0.7412 |
| B-U (HPD) | 1.0297 | 1.8060 | 0.7762 |
| Wald | 1.0316 | 1.7332 | 0.7016 |
| Fieller | 1.0659 | 1.7831 | 0.7172 |

constructing confidence intervals for the ratio of CVs of lognormal distributions with excess zeros.

Furthermore, the confidence intervals evaluation in the empirical study is coincidental with the simulation results.

## Author Contributions

**Conceptualization:** Sa-Aat Niwitpong.

**Data curation:** Noppadon Yosboonruang.

**Formal analysis:** Noppadon Yosboonruang, Sa-Aat Niwitpong, Suparat Niwitpong.

**Funding acquisition:** Sa-Aat Niwitpong.

**Investigation:** Suparat Niwitpong.

**Methodology:** Noppadon Yosboonruang, Suparat Niwitpong.

**Project administration:** Suparat Niwitpong.

**Resources:** Noppadon Yosboonruang.

**Software:** Noppadon Yosboonruang.

**Supervision:** Sa-Aat Niwitpong, Suparat Niwitpong.

**Visualization:** Sa-Aat Niwitpong, Suparat Niwitpong.

**Writing – original draft:** Noppadon Yosboonruang.

**Writing – review & editing:** Sa-Aat Niwitpong.

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
