## [Decision Letter · Decision Letter 0]

19 Jan 2022

PONE-D-21-38266Bayesian interval estimations for rainfall dispersions using the ratio of two coefficients of variation of lognormal distributions with excess zerosPLOS ONE

Dear Dr. Niwitpong,

Thank you for submitting your manuscript to PLOS ONE. After careful consideration, we feel that it has merit but does not fully meet PLOS ONE’s publication criteria as it currently stands. Therefore, we invite you to submit a revised version of the manuscript that addresses the points raised during the review process.

Specifically, reviewers raise some concerns about clarity of presentation and interpretation of the results.  A revision may be able to address these concerns.

We look forward to receiving your revised manuscript.

Kind regards,

Bryan C Daniels

Academic Editor

PLOS ONE

Journal Requirements:

"The authors gratefully acknowledge King Mongkut’s University of Technology North Bangkok. Contract no. KMUTNB-65-KNOW-09."

We note that you have provided funding information. However, funding information should not appear in the Acknowledgments section or other areas of your manuscript. We will only publish funding information present in the Funding Statement section of the online submission form. 

"The authors gratefully acknowledge King Mongkut’s University of Technology North 

Bangkok. Contract no. KMUTNB-65-KNOW-09.

Reviewers' comments:

Reviewer's Responses to Questions

**Comments to the Author**

1. Is the manuscript technically sound, and do the data support the conclusions?

Reviewer #1: No

Reviewer #2: Partly

2. Has the statistical analysis been performed appropriately and rigorously? 

Reviewer #1: Yes

Reviewer #2: Yes

3. Have the authors made all data underlying the findings in their manuscript fully available?

Reviewer #1: Yes

Reviewer #2: Yes

4. Is the manuscript presented in an intelligible fashion and written in standard English?

Reviewer #1: Yes

Reviewer #2: Yes

5. Review Comments to the Author

Reviewer #1: The authors consider Fiducial, Bayesian, Frequentist interval estimations for rainfall dispersions using the ratio of two coefficients of variation of lognormal distributions with excess zeros. The topic and application are interesting. I have a few comments given below.

1. The paper considered Fiducial, Bayesian, and Frequentist methods. Why is the title about Bayesian interval estimations?

2. Besides the implication from simulations, how would one choose prior for Bayesian methods particularly for this rainfall application? Empirical Bayes might be another option.

3. Is it reasonable to assume that non-zero $X$ follows lognormal? Nonparametric fiducial approach recently drew some attention.

4. Real data application, I am not convinced by the statement that the HPD Bayesian using the Jeffreys rule prior is the best.

The shortest interval does not mean the best as the ground truth is not known, right?

Reviewer #2: In this paper, some approaches have been proposed for the ratio of coefficients of variation in the log-normal distribution with excess zeros. The paper can e accepted for publication in Plos One. However, the English language must be improved.

6. PLOS authors have the option to publish the peer review history of their article (what does this mean?). If published, this will include your full peer review and any attached files.

Reviewer #1: No

Reviewer #2: No

---

## [Author Response · Author response to Decision Letter 0]

6 Feb 2022

Response to the Academic Editor and Reviewers

Journal: PLOS ONE

Manuscript ID: PONE-D-21-38266

Title name: Bayesian interval estimations for rainfall dispersions using the ratio of two coefficients of variation of lognormal distributions with excess zeros

Authors: Noppadon Yosboonruang, Sa-Aat Niwitpong, Suparat Niwitpong

Dear Academic Editor and Reviewers, 

 We are very grateful for your constructive comments to our manuscript. The manuscript was revised in accordance with your valuable suggestions. Below are our point-by-point replies to your comments.

Lists of correction:

Reviewer 1

1. The paper considered Fiducial, Bayesian, and Frequentist methods. Why is the title about Bayesian interval estimations?

Response: 

 We defined the topic as Bayesian interval estimations because it is a method that can apply to various problems of interest to the researchers. The prior distribution can determine as appropriate with the distribution of data. In addition, the results obtained from this study indicate that Bayesian is the best method when compared with the other methods. To avoid doubt, we changed the title to “Confidence intervals for rainfall dispersions using the ratio of two coefficients of variation of lognormal distributions with excess zeros”.

2. Besides the implication from simulations, how would one choose prior for Bayesian methods particularly for this rainfall application? Empirical Bayes might be another option.

Response: 

 According to testing the distribution of rainfall data, there was a lognormal distribution with excess zeros. Therefore, we choose the prior suitability for Bayesian by considering the values of a random variable of their posterior distributions that correspond to those for a lognormal distribution with excess zeros, see, for example, Maneerat et al. (2021a, 2021b) and Yosboonruang et al. (2021). We will find new confidence intervals for your valuable comments using the Empirical Bayes in the next study.

3. Is it reasonable to assume that non-zero follows lognormal? Nonparametric fiducial approach recently drew some attention.

Response: 

 Yes, the reason to assume the distribution of non-zero follows lognormal either from the normal Q-Q plot or the minimum AIC and BIC. For example, rainfall data has a right-skewed distribution that may correspond with various distributions; therefore, consider the distribution of such data from the normal Q-Q plot or the minimum AIC and BIC, based on an empirical example in the manuscript. The following papers show that the delta-lognormal distribution fits the rainfall data, see for example:

 Yosboonruang et al. (2021) used the minimum AIC and the lowest BIC to test the fitting of the distributions to the daily rainfall data. Then, they confirmed it with the normal Q-Q plots to show the distributions of such data.

 Maneerat et al. (2021b) checked a fit of distribution of rainfall data by a minimum AIC and showed the distribution of the datasets by histogram plots and normal Q-Q plots.

 We will study the value of your suggestion for the nonparametric fiducial approach the next time.

4. Real data application, I am not convinced by the statement that the HPD Bayesian using the Jeffreys rule prior is the best. The shortest interval does not mean the best as the ground truth is not known, right?

Response: 

 Thank you for your recommendation, we agree with your opinion. Therefore, we deleted the sentence “The results reveal that the Bayesian method using the Jeffreys rule prior based on the HPD interval had the shortest interval length from the simulation results for large sample sizes.” in the summary of an empirical example in a revised manuscript. Likewise, we adjusted the last sentence in the conclusions section on page 13/20 in a revised manuscript as “Furthermore, the confidence intervals evaluation in the empirical study is coincidental with the simulation results.”.

References

Maneerat P, Niwitpong S, Niwitpong S. Bayesian confidence intervals for a single mean and the difference between two means of delta-lognormal distributions. Commun Stat-Simul C. 2021a;50(10):2906–2934.

Maneerat P, Niwitpong S, Niwitpong S. Simultaneous confidence intervals for all pairwise comparisons of the means of delta-lognormal distributions with application to rainfall data. PLoS ONE. 2021b;16(7):e0253935.

Yosboonruang N, Niwitpong S, Niwitpong S. Simultaneous confidence intervals for all pairwise differences between the coefficients of variation of rainfall series in Thailand. PeerJ. 2021;9:e11651.

Reviewer 2

In this paper, some approaches have been proposed for the ratio of coefficients of variation in the log-normal distribution with excess zeros. The paper can be accepted for publication in PLOS ONE. However, the English language must be improved.

Response: 

 Thank you for reviewing the manuscript. The manuscript has been improved by native.

Best regards,

Noppadon Yosboonruang, Sa-Aat Niwitpong, and Suparat Niwitpong

The authors

---

## [Decision Letter · Decision Letter 1]

10 Mar 2022

Confidence intervals for rainfall dispersions using the ratio of two coefficients of variation of lognormal distributions with excess zeros

PONE-D-21-38266R1

Dear Dr. Niwitpong,

We’re pleased to inform you that your manuscript has been judged scientifically suitable for publication and will be formally accepted for publication once it meets all outstanding technical requirements.

Kind regards,

Bryan C Daniels

Academic Editor

PLOS ONE

Additional Editor Comments (optional):

Reviewers' comments:

Reviewer's Responses to Questions

**Comments to the Author**

1. If the authors have adequately addressed your comments raised in a previous round of review and you feel that this manuscript is now acceptable for publication, you may indicate that here to bypass the “Comments to the Author” section, enter your conflict of interest statement in the “Confidential to Editor” section, and submit your "Accept" recommendation.

Reviewer #1: All comments have been addressed

Reviewer #2: All comments have been addressed

2. Is the manuscript technically sound, and do the data support the conclusions?

Reviewer #1: Partly

Reviewer #2: Partly

3. Has the statistical analysis been performed appropriately and rigorously? 

Reviewer #1: Yes

Reviewer #2: Yes

4. Have the authors made all data underlying the findings in their manuscript fully available?

Reviewer #1: Yes

Reviewer #2: Yes

5. Is the manuscript presented in an intelligible fashion and written in standard English?

Reviewer #1: Yes

Reviewer #2: Yes

6. Review Comments to the Author

Reviewer #1: All comments from the last round were addressed. No further comments given in this round.

Reviewer #2: (No Response)

7. PLOS authors have the option to publish the peer review history of their article (what does this mean?). If published, this will include your full peer review and any attached files.

Reviewer #1: No

Reviewer #2: No

---

## [Editor Report · Acceptance letter]

14 Mar 2022

PONE-D-21-38266R1 

Confidence intervals for rainfall dispersions using the ratio of two coefficients of variation of lognormal distributions with excess zeros 

Dear Dr. Niwitpong:

I'm pleased to inform you that your manuscript has been deemed suitable for publication in PLOS ONE. Congratulations! Your manuscript is now with our production department. 

Kind regards, 

on behalf of

Dr. Bryan C Daniels 

Academic Editor

PLOS ONE